# WNK1 signalling regulates amino acid transport and mTORC1 activity to sustain acute myeloid leukaemia growth

Shunlei Duan [1,2,9], Karl Agger[2,9], Jan-Erik Messling[2], Koutarou Nishimura[2,3], Xuerui Han[1,2], Isabel Peña-Rømer [1], Pavel Shliaha[4], Helene Damhofer [1,2,3], Max Douglas [1], Manas Kohli [1], Akos Pal[5], Yasmin Asad [5], Aaron Van Dyke [6], Raquel Reilly[6], Robert Köchl[7], Victor L. J. Tybulewicz [7], Ronald C. Hendrickson[4], Florence I. Raynaud [5], Paolo Gallipoli [8], George Poulogiannis [1] & Kristian Helin [1,2,3] ✉

The lack of curative therapies for acute myeloid leukaemia (AML) remains an ongoing challenge despite recent advances in the understanding of the molecular basis of the disease. Here we identify the WNK1-OXSR1/STK39 pathway as a previously uncharacterised dependency in AML. We show that genetic depletion and pharmacological inhibition of WNK1 or its downstream phosphorylation targets OXSR1 and STK39 strongly reduce cell proliferation and induce apoptosis in leukaemia cells in vitro and in vivo. Furthermore, we show that the WNK1-OXSR1/STK39 pathway controls mTORC1 signalling via regulating amino acid uptake through a mechanism involving the phosphorylation of amino acid transporters, such as SLC38A2. Our findings underscore an important role of the WNK1-OXSR1/STK39 pathway in regulating amino acid uptake and driving AML progression.

Acute myeloid leukaemia (AML) is a cancer of the haematopoietic myeloid lineage, characterised by a vast molecular heterogeneity and poor prognosis[1]. Despite recent advances in understanding the underlying molecular genetics and cytogenetic alterations of AML[1–4], and the successful development of new targeted therapies for patients with specific genetic lesions, primary resistance and relapse remain a challenging issue[5,6].

With-No-lysine (K) kinase 1 (WNK1) belongs to a subfamily of four atypical serine-threonine kinases characterised by their lack of a conserved catalytic lysine in the kinase subdomain II that is crucial for ATP binding[7,8]. The WNK kinases, instead, contain an alternative lysine in subdomain I that reaches into subdomain II, thereby providing catalytic activity. Whereas WNK1 is widely expressed in

mammalian tissues, WNK2-WNK4 show more tissue-restricted expression[9].

WNK1 plays a critical role in ion transport through its effector kinases oxidative stress responsive kinase 1 (OXSR1) and STE20/SPS1-related proline-alanine-rich protein kinase (STK39/SPAK)[10]. WNK1-mediated phosphorylation and activation of the OXSR1/STK39 kinases, controls the activity of the solute carrier (SLC12) family of cation-coupled chloride co-transporters. The $Na^+/K^+$-coupled $Cl^-$ importers SLC12A1(NKCC1), SLC12A2 (NKCC2) and SLC12A3 (NCC)[11–17] are activated, whilst the $K^+$-coupled $Cl^-$ exporters SLC12A4-SLC12A7 (KCC1-KCC4) are inhibited via this phosphorylation[18].

Gain-of-function mutations increasing the expression of WNK1 have been found in patients with familial hypertension, characterised

[1]Division of Cell and Molecular Biology, The Institute of Cancer Research, Londo, UK. [2]Biotech Research and Innovation Centre, University of Copenhagen, Copenhagen, Denmark. [3]Cell Biology Program and Center for Epigenetics Research, Memorial Sloan Kettering Cancer Center, New York, NY, USA. [4]Microchemistry and Proteomics Core Facility, Memorial Sloan Kettering Cancer Center, New York, NY, USA. [5]Division of Cancer Therapeutics, The Institute of Cancer Research, London, UK. [6]Department of Chemistry & Biochemistry, Fairfield University, Fairfield, CT, USA. [7]The Francis Crick Institute, London, UK. [8]Centre for Haemato-Oncology, Barts Cancer Institute, Queen Mary University of London, London, UK. [9]These authors contributed equally: Shunlei Duan, Karl Agger. ✉e-mail: kristian.helin@icr.ac.uk

by increased salt reabsorption, and impaired K$^+$ and H$^+$ excretion in the kidney[19]. WNK1 is therefore a potential target for the treatment of hypertension, which has led to the development of several small molecule WNK1 inhibitors[20,21]. WNK1 has also been associated with a variety of other processes, including regulation of cell volume[17,22,23], sensing of molecular crowding[24], autophagy[25,26], mitosis and abscission[27], efferocytosis[28], assembly of the ER membrane protein complex[29], T and B cell activation, adhesion and migration[30–33], thymocyte development[34], and regulation of NLRP3 inflammasome activation[35]. Interestingly, several studies have suggested that WNK1 also plays an important role in cancer[36]. However, the specific contribution of WNK1 to sustain cancer cells remains unclear. In this study, we show that the WNK1-OXSR1/STK39 pathway regulates amino acid uptake and controls mTORC1 signalling, suggesting that the pathway plays an essential role in regulating cancer cell metabolism.

## Results

### WNK1 is an essential dependency in AML

To identify protein kinases that are essential for AML cells, we employed a CRISPR/Cas9–based negative selection screen with a custom protein kinase domain–focused single guide RNA (sgRNA) library targeting the 545 annotated murine protein kinases[37]. The library contained 6237 sgRNAs (3–10 per kinase), 100 positive control sgRNAs targeting known essential genes and 1000 negative control sgRNAs. The screen was performed in a mouse model of leukaemia driven by the KMT2A–MLLT3 (MLL-AF9) fusion protein[38] stably expressing Cas9. 129 different sgRNAs were significantly decreased, which targeted 37 genes (Supplementary Fig. 1a, b), including genes coding for CDK1[39], CDK9[40], ATR[41] and JAK2[42], known to be important for AML maintenance. Recently, we validated RIOK2, one of the top dropout hits from the screen, as an essential dependency in AML[43]. The recent development of small molecule inhibitors to WNK kinases[20,21] prompted us to interrogate the suitability of WNK1, another top dropout kinase hit from our screen (Supplementary Fig. 1a, b), as a potential target for AML.

To validate our initial observation, we cloned the two sgRNAs against *Wnk1* giving the strongest dropout in the screen and transduced them individually in the Cas9-expressing MLL-AF9 driven mouse leukaemia cells (hereinafter referred to as MA9 leukaemia cells) (Fig. 1a). Genetic disruption of *Wnk1* strongly suppressed the growth of MA9 leukaemia cells to a degree comparable to targeting the essential gene *Rps19* (Fig. 1b). By contrast, targeting *Wnk1* in mouse embryonic fibroblasts (MEFs) had no significant effect (Fig. 1c).

To assess if WNK1 is also required for human AML cells, we deleted *WNK1* using CRISPR/Cas9 in a series of human AML cell lines with different driver mutations. Targeting *WNK1* in all the human AML cell

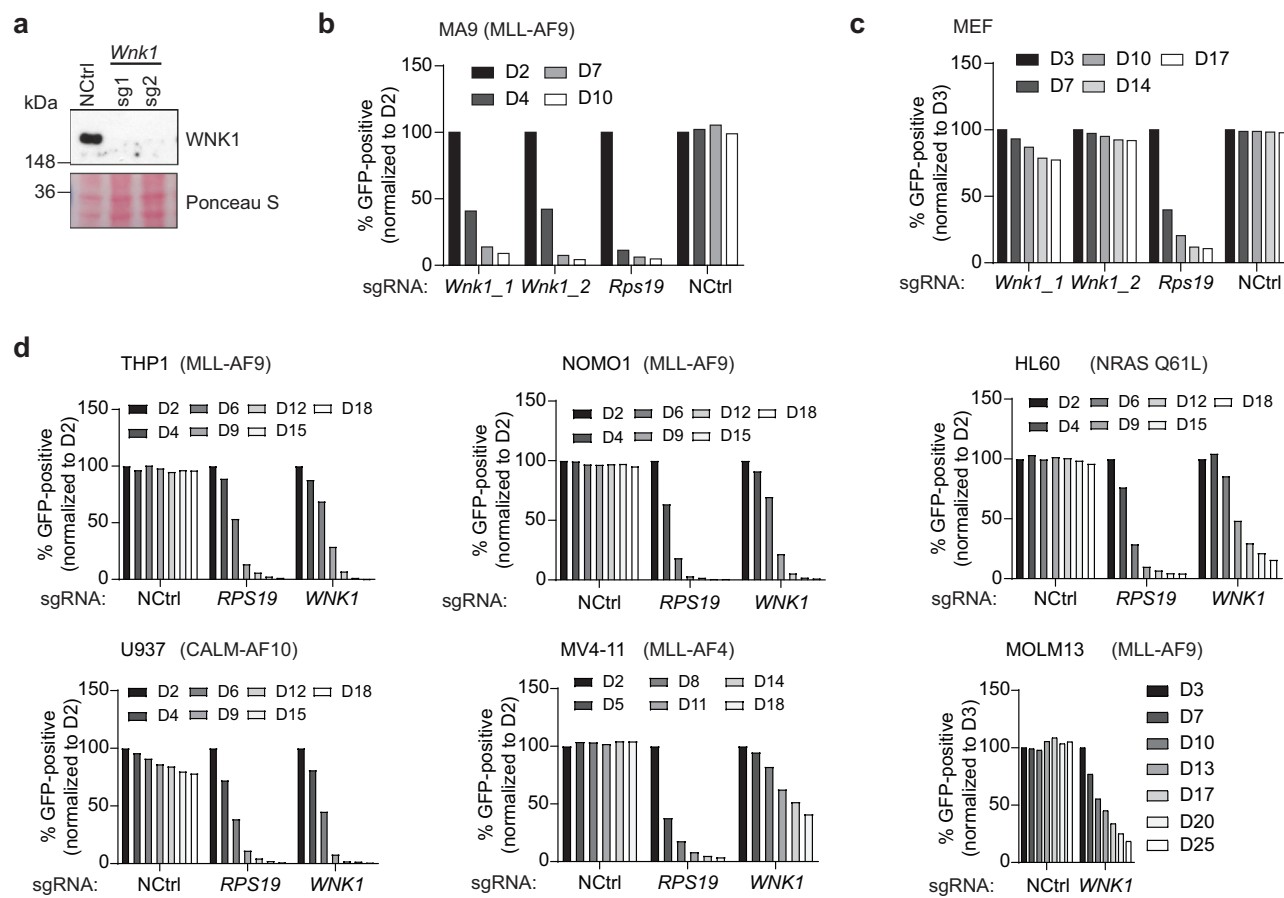

**Fig. 1 | WNK1 is an essential dependency in AML. a** Immunoblot of WNK1 in Cas9-expressing MA9 leukaemia cells expressing either a non-targeting (NCtrl) sgRNA or two different sgRNAs targeting *Wnk1*. Ponceau S staining was used as a loading control. Blots are from one representative experiment (*n* = 1). **b, c** Drop-out growth competition assays showing the relative percentage over time in days (D) of the indicated sgRNA-positive (GFP positive) Cas9-expressing MA9 leukaemia cells (**b**), mouse embryonic fibroblasts (MEFs) (**c**). An sgRNA against an essential gene (*Rps19*) was used as a positive control and a non-targeting sgRNA (NCtrl) was used as a negative control. Data shown are from one representative experiment (*n* = 1). **d** Drop-out growth competition assays showing the relative percentage over time in days of the indicated Cas9-expressing human AML cells that express a WNK1 sgRNA. An sgRNA against an essential gene (*RPS19*) was used as a positive control and a non-targeting sgRNA (NCtrl) was used as a negative control. Data shown for NOMO-1, HL-60, U937, and MOLM-13 are from one representative experiment (*n* = 1). Data shown for THP-1 and MV4-11 are from one of two independent experiments (*n* = 2). The driving oncogene is indicated for each of the AML cell lines.

lines led to impaired growth, extending our results from the mouse MA9 leukaemia cells (Fig. 1d). We then queried the DepMap database (https://depmap.org/portal/) to interrogate if tumour cell lines in general are dependent on WNK1, and showed that this dependency extends to most tumour cell lines, particularly leukaemia and multiple myeloma cell lines[31,44,45] (Supplementary Fig. 1c). Taken together, these data show that WNK1 is essential for the growth of AML cells, but importantly also for the proliferation of most cancer cell lines tested.

## WNK1 dependency in AML is mediated through OXSR1/STK39

Next, we investigated whether the two related kinases, OXSR1 and STK39, the best-characterised downstream effector kinases of WNK1, are also required for AML (Fig. 2a). Targeting OXSR1 strongly suppressed the growth of MA9 leukaemia cells (Fig. 2b, c), in which the expression of STK39 is undetectable (Supplementary Fig. 2a). The CRISPR/Cas9 screen in the MA9 leukaemia cells supported this observation (Supplementary Fig. 1a). Several results suggest that OXSR1 and STK39 are functionally redundant kinases[10,46,47]. Consistent with this, targeting of OXSR1 or STK39 alone had no detrimental effect on the proliferation of human AML cells that express both OXSR1 and STK39 (Fig. 2d, e and Supplementary Fig. 2b–h). In contrast, co-targeting of OXSR1 and STK39 had a strong inhibitory effect on the proliferation of these cells (Fig. 2d, e and Supplementary Fig. 2b–h).

WNK1 activates OXSR1 by phosphorylating a conserved threonine residue (Thr185) within its catalytic *T*-loop domain and a serine residue (Ser325) located within the non-catalytic *C*-terminal domain. Similarly, WNK1 activates STK39 by phosphorylating a conserved threonine residue (Thr233) within its catalytic T-loop domain and a serine residue (Ser373) located within the non-catalytic C-terminal domain (Fig. 2f)[10,48]. As OXSR1 and STK39 are activated by WNK1 mediated phosphorylation, we next investigated whether their kinase activity is required for AML growth. In mouse MA9 leukaemia cells, ectopic expression of an sgRNA-non-targetable wild-type human OXSR1 rescued the reduced proliferation phenotype observed by the disruption of endogenous OXSR1 (Fig. 2g, h). Ectopic expression of an sgRNA-non-targetable phosphomimetic mutant of human OXSR1 (OXSR1[T185E, S325E]) was also able to rescue the phenotype caused by the disruption of endogenous *OXSR1*, whereas ectopic expression of an sgRNA-non-targetable catalytically inactive mutant of human OXSR1 (OXSR1[D164A]) failed to do so (Fig. 2g, h). These results demonstrate that the catalytic activity of OXSR1 is required for cell proliferation in MA9 cells lacking STK39 expression and that the phosphomimetic OXSR1 mutant retains its catalytic activity.

To investigate the linearity of the WNK1-OXSR/STK39 pathway for supporting AML proliferation, we tested if expression of human OXSR1[T185E, S325E] could overcome the requirement for WNK1. OXSR1[T185E, S325E], but not OXSR1[WT] or OXSR1[D164A], maintained MA9 leukaemia cell proliferation in the absence of WNK1 (Fig. 2i). Similar results were obtained in human AML cells, where ectopic expression of either constitutively active phosphomimetic mutant of OXSR1 (OXSR1[T185E, S325E]) or constitutively active phosphomimetic mutant of STK39 (STK39[T233E, S373E]) restored cell proliferation defects caused by the disruption of endogenous *WNK1*, but neither OXSR1[WT] nor STK39[WT] were able to do so (Fig. 2j and Supplementary Fig. 2i, j). Furthermore, targeting WNK1 does not affect the expression of the OXSR1 and STK39 proteins (Supplementary Fig. 3a). Taken together, these results suggest that the role of WNK1 in regulating AML growth is mediated, at least in part, through the phosphorylation of OXSR1/STK39, as supported by this and previous studies.

## Targeting WNK1 prolongs the survival of mice with AML

To investigate if WNK1 is required for AML progression in vivo, we generated AML cell lines harbouring an *MLL-AF9* translocation by transducing c-KIT-enriched bone marrow cells from mice with a conditional allele of *Wnk1* (*Wnk1*[fl/+]), mice with a deleted allele of *Wnk1* (*Wnk1*[fl/−]), and mice with a kinase-inactive allele of *Wnk1* (*Wnk1*[fl/D386A]) with a retrovirus expressing the *MLL-AF9* oncogene (Fig. 3a). These mice expressed a tamoxifen inducible Cre from the *Rosa26* locus. Subsequent, propagation and transplantation of the MLL-AF9 transduced c-KIT[+] cells in recipient mice led to AML, and leukaemia cells from these mice were used for further analysis. Treatment of these leukaemia cells with 4-hydroxytamoxifen (OHT) led to recombination of the *Wnk1* locus and concomitant loss of WNK1 protein expression (Supplementary Fig. 3b), confirming that WNK1 is essential for the proliferation of these leukaemia cells in vitro (Fig. 3b and Supplementary Fig. 3c). Notably, OHT-resistant cells emerged after long-term treatment with OHT, most likely because of the incomplete recombination of the *Wnk1* locus (Supplementary Fig. 3d, e). Moreover, leukaemia cells expressing only the catalytically inactive WNK1 mutant showed a dramatic loss of proliferation (Fig. 3b), demonstrating the requirement for the catalytic activity of WNK1 to sustain AML proliferation. EdU labelling showed leukaemia cells with deletion of *Wnk1* or leukaemia cells only expressing catalytically inactive WNK1 became apoptotic and exhibited a drastic reduction of cells in S-phase (Fig. 3c).

To explore this further in vivo, sublethally irradiated recipient mice were reconstituted with *Wnk1*[f/+], *Wnk1*[f/−] or *Wnk1*[f/D368A] leukaemia cells and the mice were treated with tamoxifen or vehicle control (oil) starting two weeks after transplantation and treatment continued for 7 days (Fig. 3d). Induced recombination of the *Wnk1* locus resulted in increased survival for mice transplanted with *Wnk1*[f/+], *Wnk1*[f/−] or *Wnk1*[f/D368A] leukaemia cells (Fig. 3e–g), though the effect for mice transplanted with *Wnk1*[f/+] was less profound. To investigate why tamoxifen-treated mice still succumbed to leukaemia, we genotyped bone marrow isolated at disease onset from mice transplanted with *Wnk1*[f/−] leukaemia cells. The genotyping showed that the *Wnk1* locus was not completely lost in the tamoxifen-treated group, indicating that the AML in tamoxifen-treated mice may have arisen from cells in which the *Wnk1* allele was not deleted (Supplementary Fig. 3f, g). Taken together, these results demonstrate that loss of WNK1 and its catalytic activity impairs AML cell growth in vivo.

## Pharmacological inhibition of WNK1 reduces AML growth

To further assess the therapeutic potential of WNK1 for the treatment of AML, we tested two previously described selective small-molecule WNK inhibitors: the ATP-competitive WNK inhibitor WNK463 that inhibits all four WNK kinases as well as the allosteric and specific WNK1 inhibitor Compound 12[20,21]. Western blot (WB) analysis in human AML cells showed that both Compound 12 and WNK463 produced a dose-dependent inhibition of WNK1 activity as measured by the reduction of phosphorylation of the WNK kinase substrates OXSR1 and STK39 (Fig. 4a), consistent with previously reported EC50 values for this phosphorylation[20,21]. Treatment with increasing concentrations of Compound 12 and WNK463 led to a strong decrease in cell proliferation in both mouse and human AML cell lines (Fig. 4b–d and Supplementary Fig. 3h–k). We observed a slight rebound in the levels of phospho-OXSR1 and phospho-STK39, accompanied by a significant increase in the total protein levels of OXSR1 and STK39 over a 72-h treatment period with Compound 12 (Supplementary Fig. 3l), suggesting the existence of compensatory mechanisms to overcome the inhibition of WNK1 kinase activity. Similarly, we found that MLL-rearranged as well as several non-MLL-rearranged cell lines are sensitive to WNK1 inhibition both by Compound 12 and WNK463 (Fig. 4e and Supplementary Fig. 4). The relatively high IC50 values of the compounds may relate to their half-lives and biological properties. Interestingly, the two non-leukaemia cell lines tested (BJ-Tert and H226) showed five- to ten-fold higher $IC_{50}$ values compared to leukaemia cell lines (Fig. 4e).

Given these in vitro results and that Compound 12 is a selective WNK1 inhibitor, we tested its anti-leukaemia effects in vivo using a mouse MLL-AF9 leukaemia model. Despite the short half-life (1.4 h) of

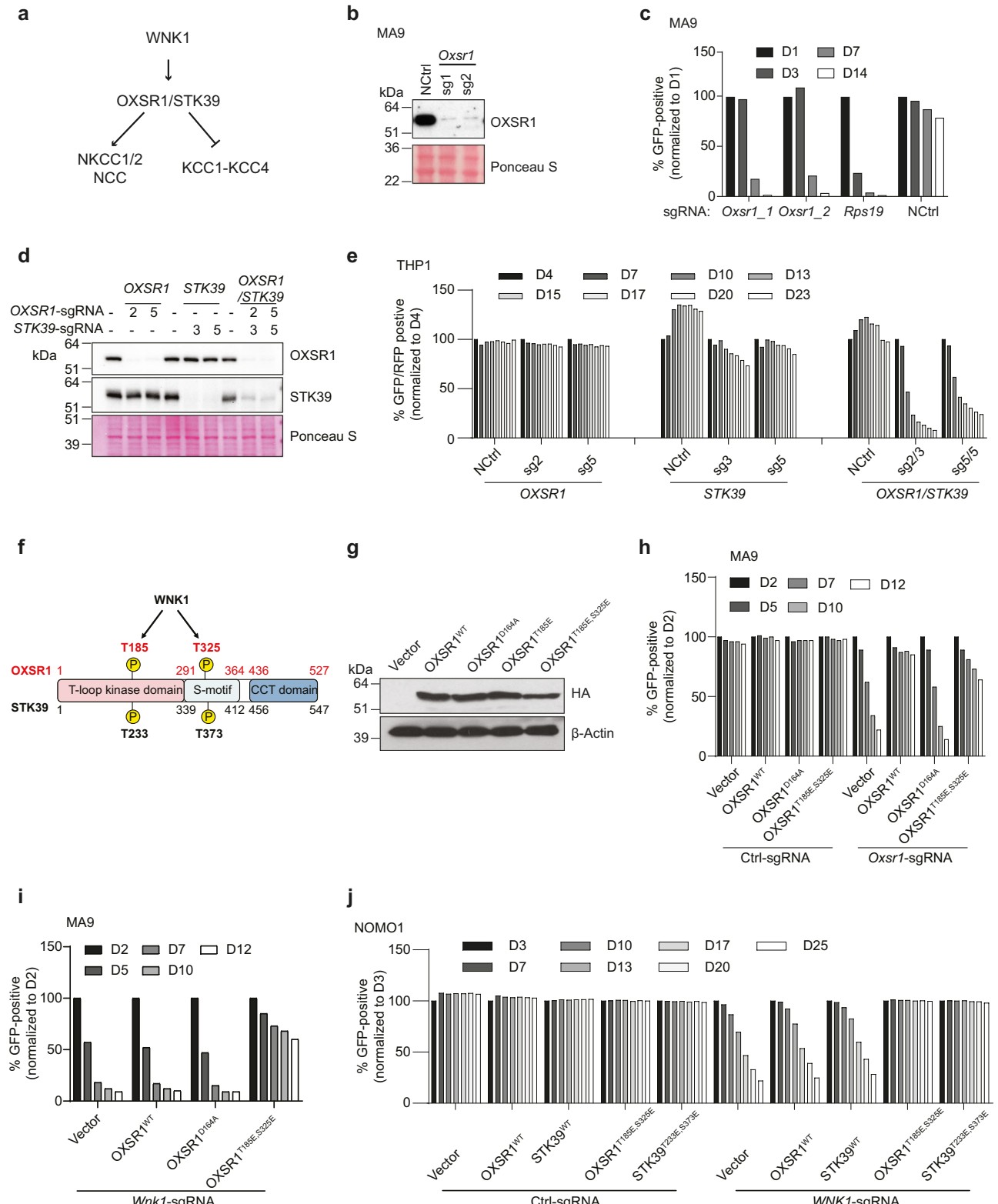

Compound 12[20], daily treatment with Compound 12 for a total period of 14 days led to a significant increase in the lifespan of the injected mice (Fig. 4f, g). These data provide pre-clinical proof-of-concept that small-molecule WNK1 kinase inhibitors may be used for the treatment of AML patients.

To further strengthen this concept, we tested the effect of WNK1 inhibition on primary human AML cells. As shown in Fig. 4h, the treatment of primary patient-derived AML samples, including both

MLL-r and non-MLL-r subtypes, with WNK463 led to reduced p-OXSR1/ p-STK39 (Fig. 4h). Notably, the IC50 of WNK463 was much lower in the primary human AML cells than in human AML cell lines and the co-cultured mesenchymal MS-5 cells (Fig. 4i). Furthermore, WNK463 treatment significantly induced apoptosis in primary AML samples (Fig. 4j). Next, we evaluated the effect of WNK1 inhibition on primary human AML cells (AML#6, see Supplementary Data 1 for molecular characterisation) transplanted into immuno-deficient NBSGW mice.

**Fig. 2 | The WNK1 dependency in AML is mediated through OXSR1/STK39.**
**a** Schematic for the WNK1-OXSR1/STK39 pathway. **b** Immunoblot of OXSR1 in Cas9-expressing MA9 leukaemia cells that express either a non-targeting sgRNA or two independent sgRNAs targeting *Oxsr1*. Ponceau S staining was used as a loading control. Blots are from one representative experiment ($n = 1$). **c** Drop-out growth competition assays showing the relative percentage over time in days of the indicated sgRNA-positive (GFP positive) Cas9-expressing MA9 leukaemia cells. An sgRNA against an essential gene (*Rps19*) was used as a positive control and a non-targeting sgRNA (NCtrl) was used as a negative control. Data shown are from a single representative experiment ($n = 1$). **d** Immunoblot of OXSR1 and STK39 in Cas9-expressing human leukaemia cells containing the indicated sgRNAs targeting *OXSR1* or *STK39*. The samples were derived from the same experiment, but different gels for OXSR1, another for STK39 were processed in parallel. Ponceau S staining was used as a loading control. Blots are from one representative experiment ($n = 1$). **e** Drop-out growth competition assays showing the relative percentage over time in days of the indicated sgRNA-positive (GFP for *OXSR1* sgRNA and RFP for *STK39* sgRNA) Cas9-expressing human THP1 AML cells. An sgRNA against an essential gene (*RPS19*) was used as a positive control and a non-targeting sgRNA

(NCtrl) was used as a negative control. Data shown are from one representative experiment ($n = 1$). **f** Schematic for WNK1-mediated phosphorylation sites on OXSR1/STK39. **g** Immunoblot using an HA antibody showing the expression of HA-tagged OXSR1[WT], OXSR1[D164A], OXSR1[18SE] and OXSR1[T18SE,S325E]. Blots are from one representative experiment ($n = 1$). The samples were derived from the same experiment, but different gels for HA, another for β-Actin were processed in parallel. **h** Drop-out growth competition assays, depicting the relative percentage over time in days of the indicated sgRNA-positive (GFP positive) Cas9-expressing MA9 leukaemia cells that ectopically express OXSR[WT], OXSR1[D164A], OXSR1[T18SE,S325E]. Data shown are from one representative experiment ($n = 1$). **i** Drop-out growth competition assays, depicting the relative percentage over time in days of the *Wnk1* sgRNA-positive (GFP positive) Cas9-expressing MA9 leukaemia cells ectopically expressing OXSR[WT], OXSR1[D164A], or OXSR1[T18SE,S325E]. Data shown are from one representative experiment ($n = 1$). **j** Drop-out growth competition assays, depicting the relative percentage over time in days of the *WNK1* sgRNA-positive (GFP positive) Cas9-expressing human leukaemia cells that ectopically express OXSR1[WT], STK39[WT], OXSR1[T18SE,S325E] or STK39[T233E,S373E]. Data shown are from one representative experiment ($n = 1$).

After 14 consecutive days of treatment with WNK463 (1.5 mg/kg, administered orally, twice daily), WNK463 treatment reduced the human AML burden in most mice bearing primary AML cells in the bone marrow (Supplementary Fig. 5a–c). However, potentially due to the high biological variability in the engraftment of the primary human AML cells[49], the reduction in the AML burden was not statistically significant.

### WNK1-OXSR1/STK39 pathway controls mTORC1 signalling

The WNK1-OXSR1/STK39 pathway regulates the NKCC1 and NKCC2 chloride co-transporters and promotes chloride flux into cells[47], particularly in the kidney cells. To understand the molecular mechanisms underlying WNK1 dependency in AML, we focused on identifying downstream phosphorylation targets of the WNK1-OXSR1/STK39 pathway that could explain its role in AML. Since neither NKCC1 nor NKCC2 have been identified as cancer dependencies in the DepMap dataset (https://depmap.org/portal/), we hypothesised that other downstream targets of this pathway may be responsible for its functional importance in AML. To identify other potential targets, we performed mass spectrometry based quantitative phosphoproteomics analysis in *Wnk1*[f/−] leukaemia cells treated with 4-hydroxytamoxifen (OHT) for 48 h to knockout *Wnk1*, or with Compound 12 for 3 h to inhibit WNK1 (Fig. 5a). To ensure that the observed changes in phosphopeptide quantitation reflect changes in site occupancy rather than changes in phosphorylated protein expression, we analysed samples without phosphoenrichment to quantify protein level changes, and with phosphoenrichment by Fe-IMAC to quantify changes in phosphopeptide levels. The phosphopeptide data was then normalised to the corresponding protein data. This approach resulted in the quantitation of 6350 proteins and 21,725 phosphosites corresponding to 9471 unique phosphopeptide sequences (Supplementary Data 2 and 3). All subsequent analyses were performed on phospho-protein data that had been normalised to protein levels. We found that 4402 unique phosphosites were regulated when comparing DMSO and Compound 12 conditions, and 3310 were regulated when comparing OHT and ethanol (EtOH) conditions (FDR < 0.05). Among these, 1156 phosphopeptides were shared, with 983 (85%) changing in the same direction in the WNK1 inhibited and depleted cells (Supplementary Fig. 5d). We focused on these shared phosphopeptides corresponding to 620 proteins, and observed that several proteins downstream of the mechanistic target of rapamycin (mTOR) complex 1 (mTORC1) signalling pathway[50–52], such as 4EBP1/2, 40S ribosomal protein S6 (RPS6), La-related protein 1 (LARP1) and DNAJC2/MPP11/ZRF1, were enriched (Fig. 5b). Gene ontology analysis of the common significantly regulated phosphoproteins in WNK1-depleted and -inhibited cells also indicated that the mTOR signalling is dependent on WNK1 activity

(Supplementary Fig. 5e). This data suggests that WNK1-OXSR1/STK39 kinases regulate mTORC1 signalling (Fig. 5c).

The mTORC1 branch of mTOR signalling pathway is a master regulator of cell growth, promoting anabolic processes while suppressing catabolic processes, and it is often deregulated in human cancers[50,53]. mTORC1 regulates multiple cellular processes, including protein synthesis, autophagy to support cell growth, proliferation, and survival, in response to a diverse set of environmental inputs ranging from levels of growth factors to energy, oxygen and/or amino acids. Inhibitors of mTOR have been demonstrated to have anti-leukaemia activity in AML[54–56]. In line with these reports, mTOR also appeared as one of the top dropout kinase hits from our CRISPR/Cas9 screen (Supplementary Fig. 1b). Thus, we decided to further explore the possibility that the WNK1-OXSR1/STK39 pathway regulates mTORC1 signalling.

To confirm the phospho-proteomics results, we directly tested the phosphorylation of the two well-characterised mTORC1 substrates, S6K1 and 4EBP1 by western blotting. WNK1 deletion or inhibition led to a rapid and dramatic decrease of S6K1 phosphorylation at Thr 389, comparable to the one observed with the mTORC1 inhibitor rapamycin (Fig. 5d and Supplementary Fig. 6a). The rapid decrease in S6K1 phosphorylation and lack of phenotypes, such as apoptosis within 4 h of WNK1 inhibition (Supplementary Fig. 6b) suggest that the effect of WNK1 on mTORC1 substrates is mediated by signalling cascades. Phosphorylation of 4EBP1 was also decreased in WNK1-depleted/inhibited cells, although to a lesser extent than S6K1 (Fig. 5d). We did not detect a significant change of 4EBP1 phosphorylation in AML cells treated with rapamycin, which is consistent with previous published results suggesting that the effect of rapamycin on 4EBP1 is cell type dependent[57,58] (Fig. 5d). Inhibition of mTORC1 with rapamycin did not affect the phosphorylation of OXSR1 at S325, indicating that the WNK1-OXSR1/STK39 pathway functions upstream of mTORC1 (Fig. 5d). However, in MEFs, deletion of *Wnk1* did not appear to affect mTORC1 signalling, (Supplementary Fig. 6c) potentially explaining that these cells are more resistant to WNK1 inhibition (Fig. 1c). Taken together, our results suggest that WNK1 is required for mTORC1 activity, whereas some cell types, such as MEFs, may compensate for this requirement.

Next, we tested whether the WNK1 requirement for mTORC1 activity is mediated through OXSR1/STK39. In *Wnk1*[f/−] leukaemia cells, ectopic expression of either constitutively active human OXSR1[T185E,S325E] or human STK39[T233E,S373E], maintained the phosphorylation of S6K1 and 4EBP1 upon OHT-induced *Wnk1* deletion (Fig. 5e). In addition, the expression of the two constitutively active kinases also partially rescued cell proliferation in *Wnk1* deleted cells (Supplementary Fig. 6d). In contrast, ectopic expression of wild-type OXSR1 and STK39 did not

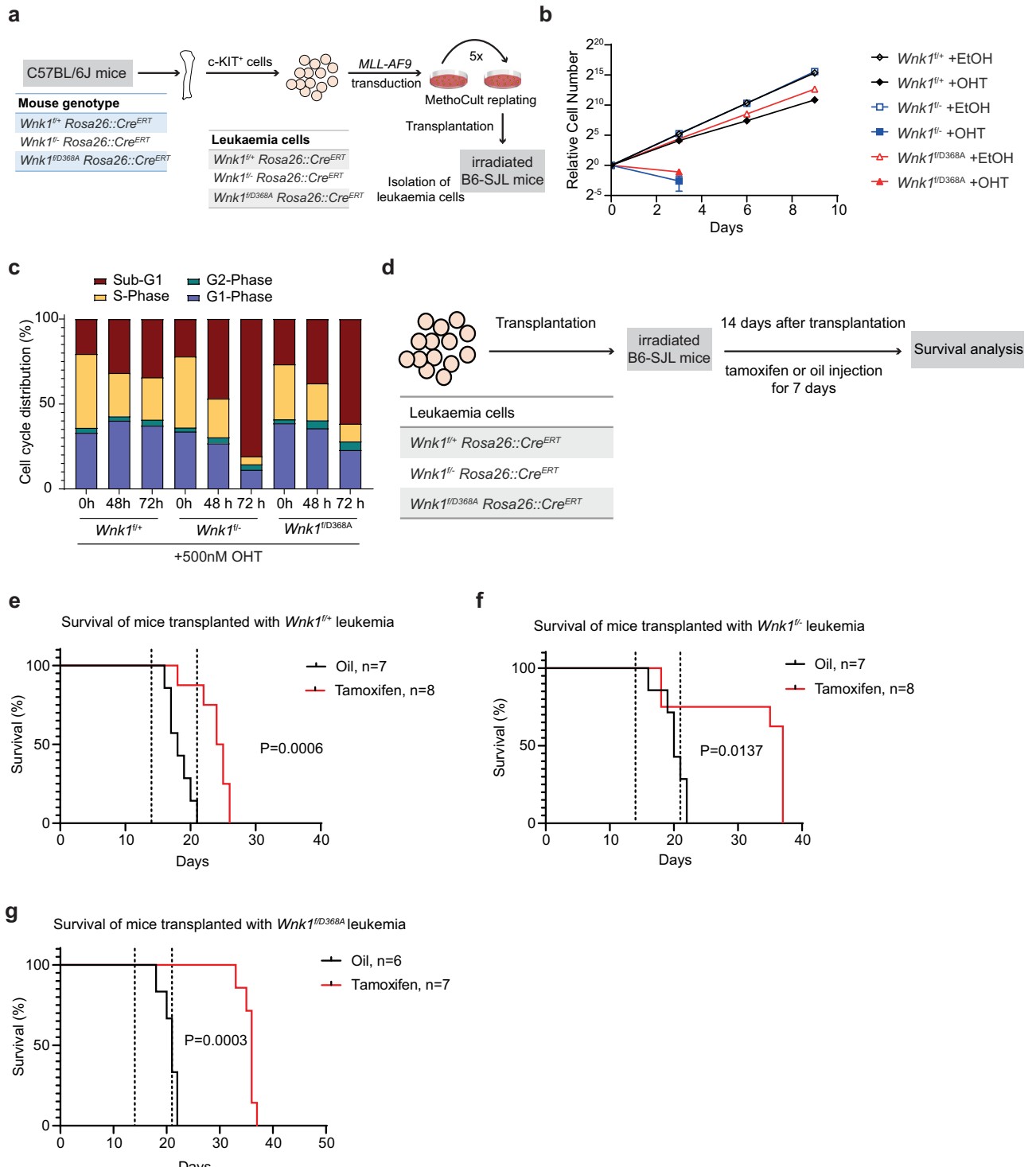

**Fig. 3 | Targeting WNK1 prolongs survival of mice with AML. a** Schematic illustration of the experimental approach employed to generate tamoxifen-inducible $Wnk1^{fl/+}$, $Wnk1^{fl/-}$ and $Wnk1^{fl/D368A}$ MLL-AF9 leukaemia cell lines. c-KIT[+] BM cells were transduced with $MLL-AF9$ oncogene and serially replated for five rounds in methylcellulose medium. The pre-leukaemia cells were transplanted into sub-lethally irradiated mice and leukaemia cells were isolated from the mice that developed leukaemia. **b** Growth curve of $Wnk1^{fl/+}$, $Wnk1^{fl/-}$ and $Wnk1^{fl/D368A}$ leukaemia cells in vitro treated with/without 500 nM OHT. Ethanol (EtOH) was used as a control. Data are presented as mean ± SD of three biological replicates ($n = 3$). **c** FACS analysis of EdU incorporation in $Wnk1^{fl/+}$, $Wnk1^{fl/-}$ and $Wnk1^{fl/D368A}$ leukaemia cells treated with 500 nM OHT for 48 and 72 h, showing a decreased S phase faction and an increased sub-G1 fraction, indicative of cell death. Data are representative of two independent experiments ($n = 2$). **d** Schematic of the experimental setup to test the requirement of WNK1 for AML in vivo. Sublethally irradiated B6.SJL recipient mice were reconstituted with $Wnk1^{fl/+}$, $Wnk1^{fl/-}$ and $Wnk1^{fl/D368A}$ leukaemia cells and the mice were treated with either tamoxifen or oil two weeks after transplantation. Mice were treated with Tamoxifen/oil for seven consecutive days and monitored for survival. **e–g** Kaplan–Meier survival curves of recipient mice transplanted with the indicated leukaemia cells. Statistical significance was calculated using a log-rank test.

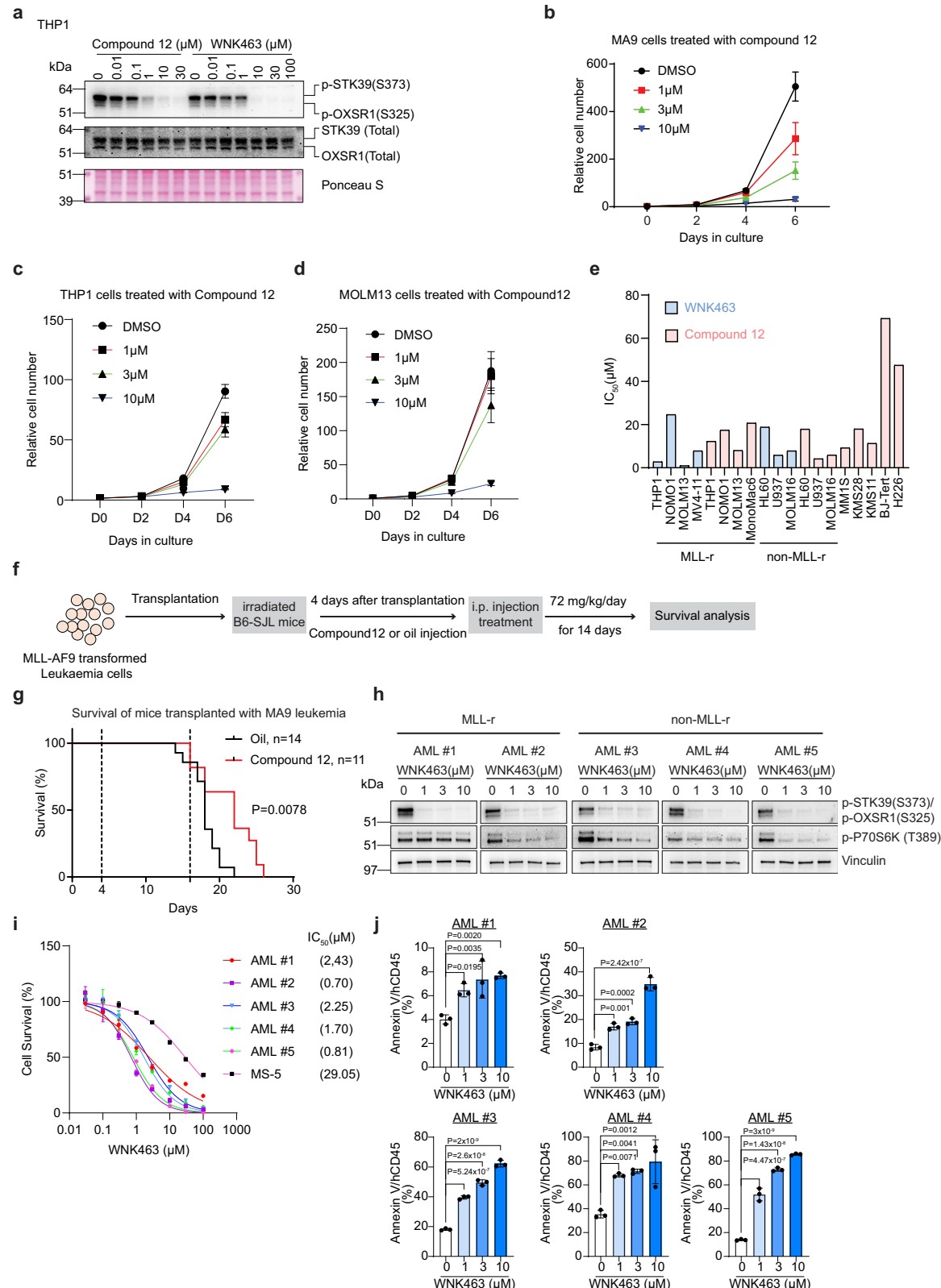

rescue mTORC1 activity or cell proliferation in *Wnk1* deleted cells (Fig. 5e and Supplementary Fig. 6d). Ectopic expression of either constitutively active human OXSR1$^{T185E,S325E}$ or human STK39$^{T233E,S373E}$ was not able to reverse the inhibitory effect of rapamycin on mTORC1 signalling, further supporting that WNK1-OXSR1/STK39 pathway functions upstream of mTORC1 (Supplementary Fig. 6e, f).

To further confirm the requirement of WNK1 signalling for mTORC1 activity, we generated MA9 leukaemia cells in which we knocked in a tag coding for mutant FKBP12 (FKBP12$^{F36V}$) in both alleles of *Wnk1* (Fig. 5f). When the resulting FKBP12$^{F36V}$-WNK1 leukaemia cells were treated with dTAG-13, WNK1 levels were undetectable within 1 h, accompanied by a significant decrease of S6K1 phosphorylation (Fig. 5f).

**Fig. 4 | Pharmacological inhibition of WNK1 reduces AML growth. a** WNK1 inhibitors elicit a dose–dependent decrease in human OXSR1 and STK39 phosphorylation. Western blot analysis of phospho-OXSR1(S325)/phospho-STK39(S373) and total OXSR1 and STK39 in human THP1 lysates treated with WNK463 or Compound 12 at the indicated concentrations. Ponceau S staining served as a loading control. Blots are from one representative experiment ($n = 1$). The samples were derived from the same experiment, but different gels for phospho-OXSR1(S325)/phospho-STK39(S373), another for OXSR1/STK39 (total) were processed in parallel. **b–d** Growth curve of MA9 (**b**), human THP1 (**c**) and human MOLM13 (**d**) leukaemia cells treated with Compound 12 at the indicated concentrations. Data are presented as mean ± SD of three independent experiments ($n = 3$). **e** IC50 values of WNK1 inhibitors for a panel of indicated cancer lines. **f** Schematic of WNK inhibitor treatment strategy. Sublethally irradiated B6.SJL recipient mice were transplanted with MA9 leukaemia cells, and the mice were intraperitoneally injected with Compound 12 (72 mg/kg) on day 4 after transplantation. The treatment with Compound12 was sustained for a period of 14 days. **g** Kaplan–Meier survival curves of recipient mice transplanted with MA9 leukaemia cells receiving oil or Compound12 treatment. Statistical significance was calculated using a log-rank test. **h** Western blot analysis of $p$-OXSR1, $p$-STK39 and $p$-S6K in AML patient samples treated with WNK463. AML #1 and AML #2 are MLL-rearranged, while AML #3, AML #4, and AML #5 are MLL-non-rearranged. Patient samples were treated with WNK463 at 1, 3, or 10 $\mu$M for 2 h. Blots for each patient sample are from one representative experiment ($n = 1$). The samples were derived from the same experiment, but different gels for $p$-STK39(S373)/$p$-OXSR1(S325) and Vinculin, another for $p$-P70S6K (T389) were processed in parallel. Vinculin served as a loading control. **i** WNK463 dose–response curves for a panel of primary AML samples. Data are presented as mean ± S.D. ($n = 3$). The IC$_{50}$ for each sample is shown in brackets. **j** Percentage of Annexin V-positive cells among human hCD45-positive cells in a panel of human primary AML samples following treatment with WNK463 at the indicated concentrations. Analysis of AML#1, AML#2, and AML#3 was performed on day 3 after treatment with WNK463, whereas AML#4 and AML#5 were analysed on day 7. Data are presented as mean ± s.d. ($n = 3$). Statistical analysis was performed using one-way ANOVA followed by Dunnett's post-hoc test to compare each WNK463 concentration to the untreated control.

In line with the known role of mTORC1 as a major driver of growth and protein synthesis[59], we found that genetic depletion or pharmacologic inhibition of WNK1 or mTOR resulted in a significant decrease in protein synthesis (Fig. 5g and Supplementary Fig. 7a, b), and cell death (Fig. 3c and Supplementary Fig. 7c–f). Taken together, these data demonstrate that an active WNK1-OXSR1/STK39 pathway promotes mTORC1 activity.

**WNK1-OXSR1/STK39 pathway regulates amino acid transport**
Intracellular availability of growth factors, energy, and amino acids regulates the activation state of mTORC1 (Supplementary Fig. 7g)[50,53]. Growth factors, via PI3K and AKT, inhibit the TSC complex and PRAS40, promoting mTORC1 activity[60,61]. Energy starvation activates AMPK, which phosphorylates Raptor and TSC2 to inhibit mTORC1[62,63]. Amino acid deficiency suppresses mTORC1 through the GATOR2-GATOR1-RAGA/B GTPase cascade[64].

To define the molecular mechanisms through which the WNK1-OXSR1/STK39 pathway regulates mTORC1 signalling, we explored how WNK1 activity affects upstream signals that regulate mTORC1 (Supplementary Fig. 7g). The lack of inhibitory changes in AKT phosphorylation at Thr308 and Ser473 following WNK1 inhibition (Supplementary Fig. 8a, b) suggested, at least in part, that WNK1 inhibition does not suppress growth factor-mediated activation of mTORC1 signalling. Similarly, we did not detect any increase in AMPK signalling, monitored by phosphorylation of AMPK at Thr172[65] and the AMPK target Beclin-1 at Ser 93[66] in response to WNK1 inhibition (Supplementary Fig. 8a, c), and it is therefore unlikely that WNK1 inhibition suppresses mTORC1 activity through AMPK-mediated inhibition of RAPTOR. Consistent with these findings, the knockout of either PRAS40 or AMPK did not reverse the decrease in mTORC1 activity in cells treated with Compound 12 (Supplementary Fig. 8d, e). In contrast, depleting NPRL2, the catalytic subunit of GATOR1, restored mTORC1 signalling upon WNK1 inhibition (Fig. 6a). However, NPRL2 depletion did not rescue cell growth following WNK1 depletion (Supplementary Fig. 8f). Together, these data indicate that WNK1 inhibition may reduce intracellular amino acid levels, as NPRL2 depletion can restore mTORC1 signalling but cannot compensate for the amino acid deficiency that impairs cell proliferation upon WNK1 inhibition/depletion.

To test if WNK1 depletion/inhibition leads to a decrease in cellular amino acid concentrations, we performed targeted metabolite analysis of amino acids (Fig. 6b). Strikingly, we observed a significant decrease in the concentration of several amino acids, including Asp, Glu, Gly and Pro in both WNK1-inhibited and depleted cells, and additionally Ser, Gln, Ala and Thr in WNK1-inhibited cells (Fig. 6b and Supplementary Data 4). A bioluminescent assay for measuring intracellular glutamine/glutamate showed that intracellular glutamine/glutamate levels were dramatically reduced upon depletion or inhibition of WNK1 (Fig. 6c, d). In addition, we detected significantly decreased uptake of glutamine in cells treated with Compound 12 (Fig. 6e), suggesting that WNK1 signalling regulates the transport of amino acids across the cellular membrane. Furthermore, we performed experiments to test the requirement for WNK1 for the activation of mTORC1 signalling following amino acid starvation. As shown in Supplementary Fig. 8g, mTORC1 activity was rescued by the addition of Leu or Gln following amino acid starvation in WNK1-uninhibited cells. However, in WNK1-inhibited cells, the addition of Leu or Gln failed to restore mTORC1 activity. This result further supports the notion that WNK1 regulates amino acid uptake and, thereby mTORC1 activity.

Next, we sought to identify a molecular mechanism that could explain why WNK1 inhibition/depletion led to a decrease in intracellular concentrations and uptake of amino acids. Amino acid uptake across the cellular membrane is mediated by amino acid transporters of the SLC superfamily[67]. Loss of function of amino acid transporters, such as SLC1A5, SLC7A5, SLC38A2 results in decreased amino acid uptake and impaired mTORC1 signalling[68–71].

Evaluation of our phosphoproteomics data showed that the phosphorylation of several amino acid transporters, including SLC38A2, SLC38A1, SLC1A5, and SLC7A1 were downregulated in WNK1 inhibited/depleted cells (Supplementary Data 5), indicating that their phosphorylation states are dependent on WNK1 signalling. We focussed our further analysis on SLC38A2, because it showed the biggest decrease in phosphorylation upon WNK1 inhibition/depletion, and it is known to contribute to the transport of most of the amino acids (Gln, Gly, Ala, Ser, Pro, Thr, Asn, and His[67,72]) that we observed to decrease upon WNK1 inhibition/depletion. Motif analysis showed that SLC38A2 contains an OXSR1/STK39 binding motif within its N-terminus cytoplasmic part of the protein[73]. Consistent with this, we observed a physical interaction between SLC38A2 and OXSR1 in cells co-expressing HA-tagged OXSR1 and V5-tagged SLC38A2 (Fig. 6f, g). To determine if SLC38A2 is indeed an OXSR1 substrate, we performed an in vitro kinase assay using recombinant proteins, which showed that OXSR1 can directly phosphorylate the N-terminus of SLC38A2 (Fig. 6h). Thus, OXSR1 can directly bind to and phosphorylate SLC38A2, and its phosphorylation is dependent on the activity of the WNK1-OXSR1/STK39 pathway, suggesting that the pathway potentially regulates the amino acid transport activity of SLC38A2. To test this further by genetic complementation, we first determined whether SLC38A2 is required for the proliferation of MA9 leukaemia cells. However, since the deletion of *SLC38A2* only resulted in modest growth defects in AML cells (Supplementary Fig. 8h), presumably due to compensatory mechanisms among the amino acid transporters[72,74], we were unable to perform the complementation analysis. Taken together, our results show that the WNK1-OXSR1/STK39 pathway is regulating the

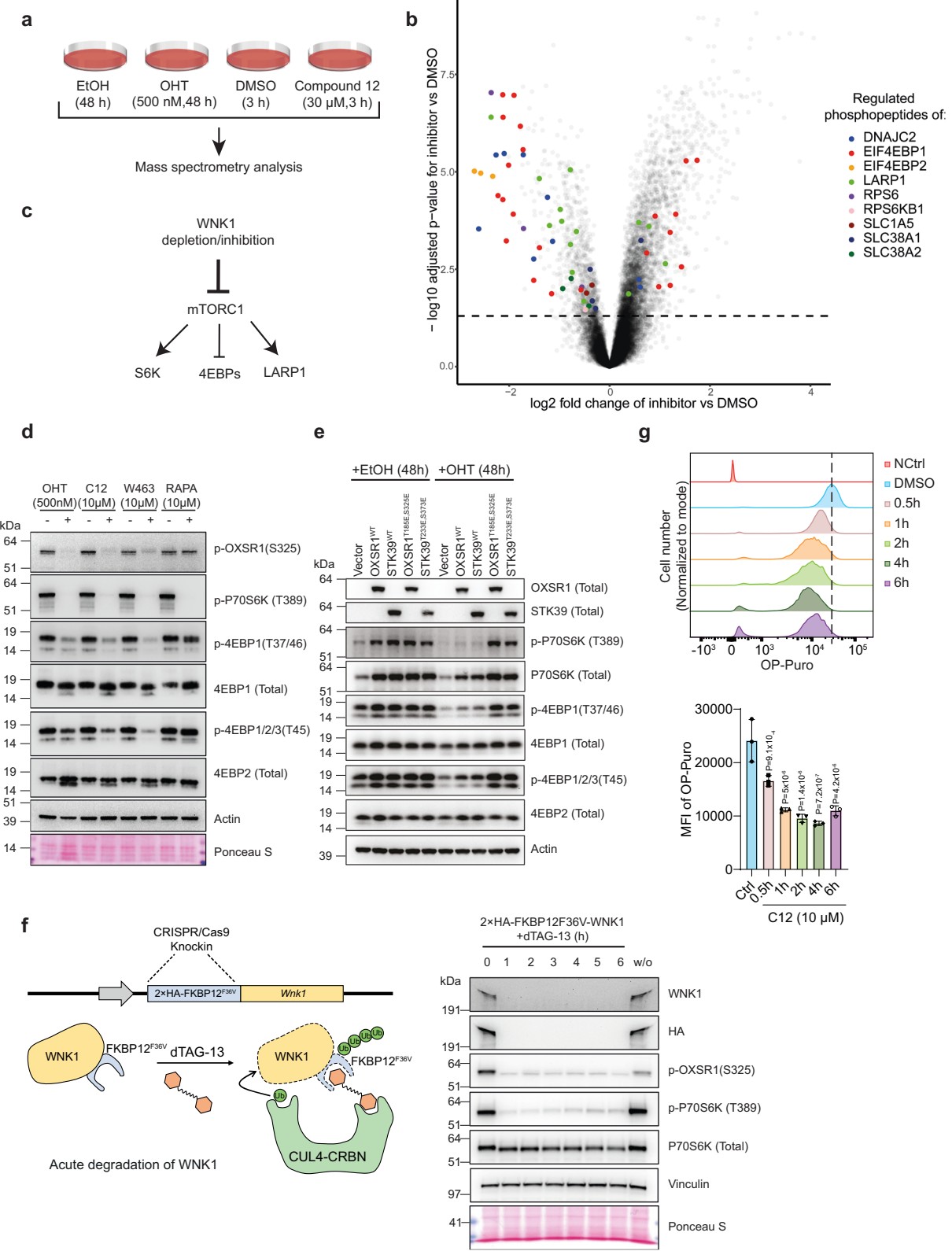

phosphorylation of several amino acid transporters, providing a potential mechanism for how this pathway controls amino acid uptake.

## Discussion

Primary resistance and relapse to treatment remain major obstacles for improving the survival outcomes of patients with AML. Here, we demonstrate that genetic or pharmacological inhibition of the WNK1-

OSXR1/STK39 pathway leads to strong suppression of AML growth in vitro and in vivo. Mechanistically, we show that targeting the WNK1-OXSR1/STK39 pathway leads to a decrease in amino acid uptake and intracellular amino acids levels, resulting in loss of mTORC1 activity, reduced protein synthesis and induction of apoptosis. Our findings also suggest that the WNK1-OXSR1/STK39 pathway is directly involved in regulating amino acid transport through

**Fig. 5 | WNK1-OXSR1/STK39 pathway controls mTORC1 signalling.**
**a** Experimental workflow for the phosphoproteome analysis. **b** Volcano plot showing changes in the phosphoproteome in MA9 leukaemia cells treated with 30 μM Compound 12 for 3 h. Significantly regulated phosphosites of mTORC1 downstream targets are highlighted in colour (FDR < 0.05). Statistical analysis was performed using the MSstatsTMT package with two-sided tests and Benjamini-Hochberg correction for multiple comparisons. Adjusted *p*-values (FDR < 0.05) were used to determine significance. **c** Schematic depicting WNK1 depletion/inhibition leads to inhibition of mTORC1 signalling. **d** Immunoblot analysis of the indicated mTORC1 downstream proteins of *Wnk1*[fl/−] MA9 leukaemia cells in the presence of OHT (500 nM, 48 h), Compound 12 (C12) (10 μM, 1 h), WNK463(W463) (10 μM, 1 h), Rapamycin (RAPA) (10 μM, 1 h). Blots are representative of at least three independent experiments (*n* > 3). The samples were derived from the same experiment, but different gels for p-OXSR1(S325) and p-4EBP1(T37/46), another for p-P70S6K (T389) and p-4EBP1/2/3(T45), another for Actin and 4EBP1 (Total), and another for 4EBP2 (Total) were processed in parallel. Ponceau S staining and Actin served as loading controls. **e** Immunoblot analysis of the indicated mTORC1 downstream proteins of *Wnk1*[fl/−] MA9 leukaemia cells ectopically expressing

OXSR1[WT], STK39[WT], OXSR1[T18SE, S325E] or STK39[T233E, S373E]. The cells were treated with OHT (500 nM) for 48 h. Blots are representative of two independent experiments (*n* = 2). The samples were derived from the same experiment, but different gels for OXSR1 (Total) and p-4EBP1(T37/46), another for STK39 (Total) and 4EBP1 (Total), another for *p*-P70S6K (T389) and p-4EBP1/2/3(T45), another for P70S6K (Total) and 4EBP2 (Total), and another for Actin were processed in parallel. Actin served as a loading control. **f** Left, schematic of the degron system for the targeted degradation of WNK1 in MA9 leukaemia cells. Right, immunoblot analysis of indicated proteins at the indicated times after treatment with 500 nM dTAG-13. Blots are representative of three independent WNK1-degron clones (*n* = 3). The samples were derived from the same experiment, but different gels for WNK1 and *p*-OXSR1(S325), another for HA and *p*-P70S6K (T389), another for P70S6K (Total) and Vinculin were processed in parallel. Ponceau S staining and Vinculin served as loading controls. **g** Protein synthesis rates as measured by incorporation of OP-puro in *Wnk1*[fl/−] MA9 leukaemia cells treated with 10 μM Compound 12 at the indicated time points. Error bars represent mean ± SD from three biological replicates (*n* = 3). Statistical analysis was performed using one-way ANOVA followed by Dunnett's post-hoc test to compare each Compound 12-treated time point to the untreated control.

the phosphorylation of SLC38A2 as well as other amino acid transporters (Fig. 6i).

mTOR signalling remains a appealing target for cancer treatment[75,76], including in leukaemia[54–56]. Amino acid sensing plays a crucial role in activating mTORC1, and targeting amino acid metabolism—either alone or in combination with mTOR inhibition—has recently been suggested as a promising strategy for cancer therapy[77,78]. Our results suggest a strategy to suppress mTORC1 signalling in AML by inhibiting WNK1-OXSR1/STK39 pathway. We demonstrated that the pathway controls mTORC1 signalling via the amino acid sensing pathway and targeting WNK1-OXSR1/STK39 pathway leads to a decrease of amino acid pools. Multiple amino acid transporters that regulate amino acid availability and function upstream of mTORC1 are required for cancer growth, such as the well-studied SLC1A5[68,70], SLC7A5[68,79], and SLC38A2[70,71,80], which are upregulated in many cancers[81]. The modest effects on AML cell growth by targeting SLC38A2 alone could be ascribed to functional redundancy with other amino acid transporters that also show WNK1-dependent phosphorylation, such as SLC38A1 and SLC1A5, which have overlapping substrate specificity with SLC38A2[67,81,82]. Currently, there is no biochemical or physiological evidence to demonstrate whether phosphorylation plays a role in the activity of SLC38A2. However, studies have shown that hyperosmotic stress, which activates the WNK1-OXSR1/STK39 pathway, increases the transport activity of SLC38A2, as measured by the uptake of its substrate analogue, α-(methylamino)isobutyric acid[83]. Our results showing that SLC38A2 is directly phosphorylated by OXSR1, suggest a potential common mechanism for how amino acid transporters are regulated by the WNK1-OXSR1/STK39 pathway. Nonetheless, further investigation is required to systematically evaluate how phosphorylation affects the function of the amino acid transporters.

Glutamine is a well-studied amino acid that regulates mTORC1 and plays a crucial role in the metabolism and growth of tumours. It facilitates the production of energy through glutaminolysis and serves as an intermediate metabolite for numerous metabolic processes, including acting as an exchange substrate for amino acid transporters. Numerous studies have demonstrated that cancer cells necessitate glutamine metabolism for its proliferation, including AML[84–86]. An increasing number of studies have shown that amino acid transporters are required for tumour growth and suggest targeting amino acid transporters as a potential therapeutic strategy[67]. Our results demonstrate that WNK1-OXSR1/STK39 pathway is involved in regulating the uptake of glutamine and other amino acids. Therefore, targeting WNK1-OXSR1/STK39 pathway provides a potential strategy for

targeting amino acid metabolism in AML therapy and could be leveraged for treating other types of cancer as well.

Despite its relatively low potency and short half-life, Compound 12 showed promising anti-tumour effectiveness and low side effects, indicating a great potential for developing or optimising WNK1 inhibitory compounds with enhanced bioavailability, stability, and potency to treat patients with AML and other cancers dependent on this pathway. We acknowledge that our study lacks an extensive evaluation of WNK1 inhibitors in a broader panel of human PDX models. While our findings provide compelling mechanistic insights and preliminary evidence of therapeutic potential, further validation in diverse and clinically relevant in vivo models is essential to fully assess the translational relevance of WNK1 targeting in AML. Such studies would help determine the consistency of response across genetically heterogeneous AML subtypes and better inform potential biomarkers of sensitivity or resistance to WNK1 inhibition.

## Methods
### Cloning
The U6-sgRNA-SFFV-puro-P2A-EGFP and pLKO5-sgRNA-EFS-tRFP657 vectors were utilised for sgRNA cloning. The U6-sgRNA-SFFV-puro-P2A-EGFP vector was generated by substituting SpCas9 open reading frame with a puromycin resistance cassette from the pL-CRISPR.SFFV.GFP plasmid (Addgene, 57827). The pLKO5-sgRNA-EFS-tRFP657 vector was a gift from Benjamin Ebert (Addgene, 57824). The sgRNA sequences were either directly taken from the mouse kinome CRISPR screen library[43] or designed using the sgRNA designer tool (https://portals.broadinstitute.org/gppx/crispick/public). The sgRNA sequences are listed in Supplementary Data 6. The cDNAs for full-length human OXSR1 and STK39 were cloned into the pLEX_307 vector (Addgene, 41392) with a *C*-terminal HA tag using gateway technology. The OXSR1 and STK39 mutations were generated using the site-directed mutagenesis (Q5 Site-Directed Mutagenesis Kit, NEB, E0554S). A DNA fragment for human SLC38A2 (1–75 aa) was synthesised from Integrated DNA Technologies UK and cloned into pET30a with a *C*-terminal GST tag.

### Reagents and antibodies
Compound 12 was synthesised according to a previously published protocol[20]. The purity of Compound 12 was determined to be >95% by HPLC, and its identity was confirmed by mass spectrometry. 4-Hydroxytamoxifen (OHT) (H7904) were purchased from Sigma, Rapamycin (S1039) from Selleck Chemicals, WNK463 (HY-100626) from MedChemExpress. For in vitro experiments, the compounds were dissolved in DMSO, except for OHT, which was dissolved in EtOH. For in vivo experiments, Compound 12 was dissolved and stored in

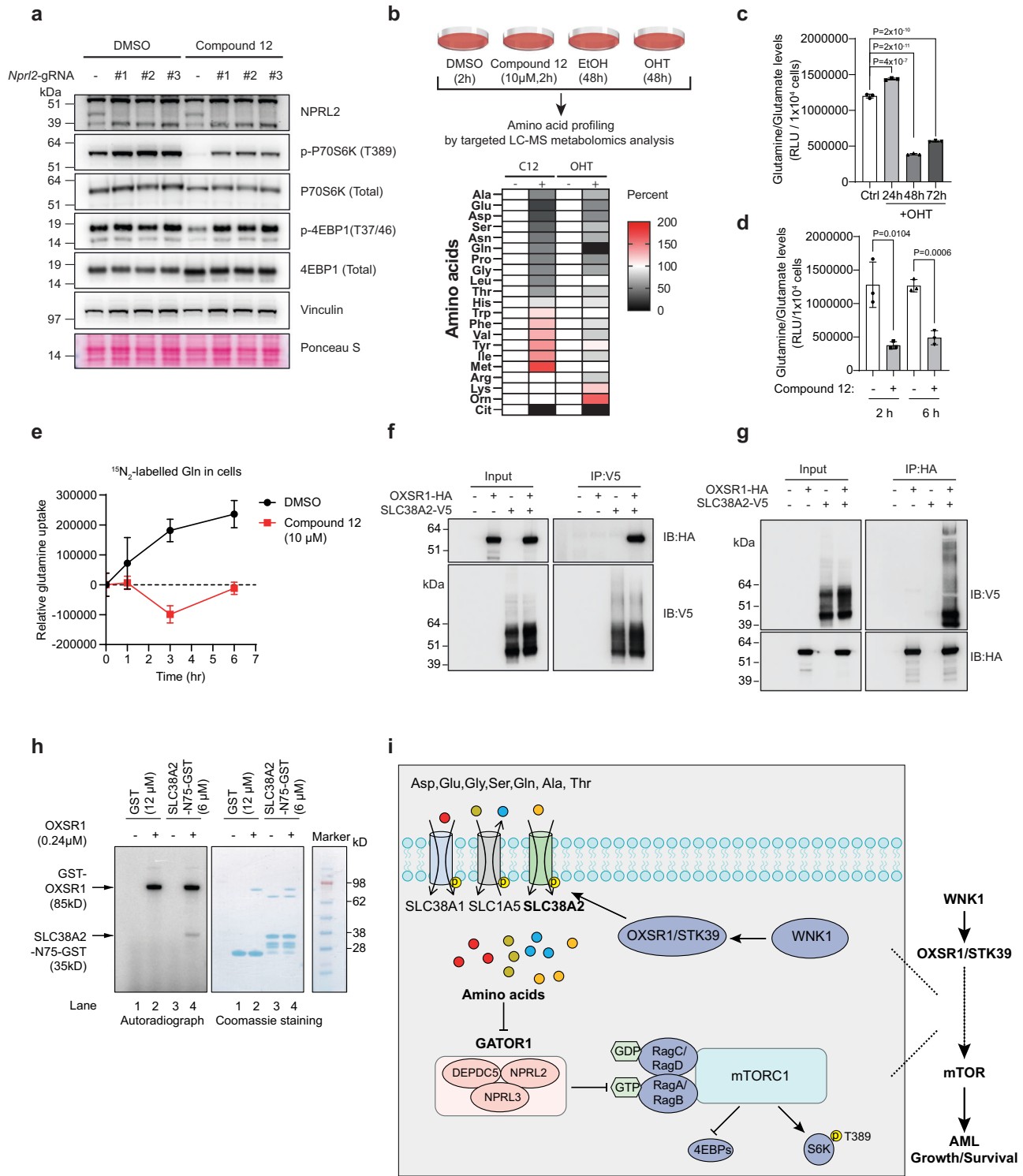

sterile corn oil; Tamoxifen (Sigma-Aldrich T5648-1G) was dissolved in cor oil at a concentration of 20 mg/ml by shaking overnight at 37 °C. Once dissolved, it was stored at 4 °C for the duration of injections. Primary and secondary antibodies used for immunoblotting and FACS are listed in Supplementary Data 7.

## Cell culture

Mouse MLL-AF9 leukaemia cells were cultured in RPMI 1640 medium (Thermo Fisher Scientific, 61870036), supplemented with 20% heat-inactivated FBS (Gibco,10500064), 20% conditioned medium from a homemade IL-3-secreting cell line and 1× penicillin/streptomycin

(Thermo Fisher Scientific, 15070063). Human THP1, NOMO1, HL60 and U937 leukaemia cells were grown in RPMI 1640 medium (Thermo Fisher Scientific, 61,870,036) containing 10% heat-inactivated FBS (Gibco, 10500064) and 1× penicillin/streptomycin (Thermo Fisher Scientific, 15070063). MV4-11, MOLM13 were grown in RPMI1640 medium with 20% heat-inactivated FBS (Gibco, 10500064) and 1× penicillin/strepto-mycin (Thermo Fisher Scientific, 15070063). HEK239T, BJ-Tert, U2OS and MEF cells were maintained in DMEM medium (Thermo Fisher Scientific, 10566016) supplemented with 10 % heat-inactivated FBS (Gibco,10500064) and 1× penicillin/streptomycin (Thermo Fisher Scientific, 15070063). All cells were maintained at 37 °C in a humidified

**Fig. 6 | WNK1-OXSR1/STK39 pathway regulates amino acid transport.**
**a** Immunoblot analysis of indicated proteins in *Wnk1*[fl/−] MA9 leukaemia cells knocking out *Nprl2* with three individual sgRNAs. The cells were treated with 10 μM Compound 12 or DMSO for 1 h. Blots are representative of at least three independent experiments (*n* > 3). The samples were derived from the same experiment, but different gels for NPRL2 and Vinculin, another for *p*-P70S6K (T389) and p-4EBP1(T37/46), another for P70S6K (Total) and 4EBP1 (Total) were processed in parallel. Ponceau S staining and Vinculin served as loading controls. **b** Schematic depicting the workflow of amino acid profiling by targeted LC-MS metabolomics analysis and relative amino acid abundance shown as a heat map. Percent changes are relative to cells without 10 μM Compound 12 (C12) or 500 nM OHT treatment. **c–d** Glutamine/Glutamate levels as measured by a bioluminescent assay in *Wnk1*[fl/−] MA9 leukaemia cells treated with 500 nM OHT (**c**) or Compound 12 (**d**) at the indicated time points. Error bars represent mean ± SD from three biological replicates (*n* = 3). Statistical analysis was performed using one-way ANOVA followed by Dunnett's post-hoc test to compare each OHT-treated time point to the untreated

control (Ctrl) in (**c**). A two-sided Student's *t* test was used to compare each Compound 12-treated time point to the untreated controls in (**d**). **e** Examination of glutamine uptake in *Wnk1*[fl/−] MA9 leukaemia cells at the indicated times after treatment with 10 μM Compound 12. Error bars represent mean ± SD from five biological replicates (*n* = 5). **f, g** Interaction of SLC38A2 with OXSR1. 293 T cells expressing the indicated HA/V5-tagged proteins were lysed and subjected to HA/V5 immunoprecipitation followed by immunoblotting for the indicated proteins. Blots are representative of two independent experiments (*n* = 2). The samples were derived from the same experiment, but different gels for HA, another for V5 were processed in parallel in **f** and **g**. **h** OXSR1 phosphorylates SLC38A2. Kinase assay was performed by incubating the indicated recombinant OXSR1, purified SLC38A2 (1–75 aa), and [γ−32P]-ATP in kinase reaction buffer at 30 °C for 1 h. Data are representative of two independent experiments (*n* = 2). The samples were derived from the same experiment, but different gels for autoradiography, and another for Coomassie staining were processed in parallel. **i** Model for the role of the WNK1-OXSR1/STK39 pathway in regulating amino acid uptake and mTORC1 signalling.

atmosphere with 5% $CO_2$. The cell lines were regularly tested for mycoplasma contamination and confirmed to be negative.

### Drop-out growth competitive assay
Cells expressing Cas9 were transduced with lentivirus containing indicated sgRNAs and analysed for GFP or RFP expression 48 or 72 h post-infection. The percentage of cells expressing the indicated sgRNAs (GFP-positive or RFP-positive) was monitored over time by flow cytometry and normalised to the initial time point.

### Virus production and lentiviral transduction
Lentivirus was produced in HEK293T cells using either a standard calcium phosphate method[87] or 1 mg/ml polyethylenimine (PEI) reagent (Polysciences, 23966) at a ratio of DNA (μg) to PEI (μl) of 1:2.5. The virus containing medium was collected 48 h after transfection, filtered through a 0.45-μm filter, and stored at −80 °C. To transduce leukaemia cells, viral supernatant was first added to RetroNectin (Takara, T100B) reagent-coated plates and centrifuged at 2000 g for 2 h. Leukaemia cells were then spun onto the coated plates, and adherent cells were infected by direct incubation with viral supernatants for 48 h in the presence of 10 μg/ml polybrene (Merck Millipore, TR-1003-G). Transduced cells were then washed with PBS and selected or analysed 48 h after transduction.

### CRISPR/Cas9–mediated kinome-wide screening
A custom sgRNA library targeting the kinase domain of 545 annotated murine protein kinases was designed and cloned into the U6-sgRNA-SFFV-puro-P2A-EGFP vector as previously described[43,88]. Briefly, an oligonucleotide pool consisting of 6237 sgRNAs targeting 545 kinases (3–10 per kinase), 100 positive control sgRNAs and 1000 negative control sgRNAs, was synthesised via microarray (CustomArray, Inc.). The oligonucleotide pool was amplified through PCR, cloned into the U6-sgRNA-SFFV-puro-P2A-EGFP vector, and subsequently verified by next-generation sequencing (NGS) for sequence confirmation. Virus containing the kinome library was produced in HEK293T cells, pre-titrated to obtain ~30% multiplicity of infection and used to transduce duplicate cultures of MA9 leukaemia cells. Each replicate screen culture was calculated to achieve a minimum of 1000 × the number of sgRNA constructs in the library. The infected cultures were selected using puromycin (2 μg/ml) commencing 48 h after transduction. The genomic DNA of the screened cells collected at day 2 and day 12 after transduction were PCR-amplified and subjected to NGS using a NextSeq 550 (Illumina). The sequencing results were analysed using DEseq2[89], and sgRNAs that exhibited depletion or enrichment at the end-point were identified as described previously[88].

### Generation of knockout cell lines
To generate knockout cell lines, cells were first selected for stable expression of Cas9 (Addgene,52962) using 5 μg/ml blasticidin. Then, these cells were transduced with and selected for stable expression of the indicated sgRNAs using 2 μg/ml puromycin. After 7 days, the pooled cells were collected for experiments or immunoblotting.

### Generation of dTAG cell lines
Generation of the WNK1-degron cell line was performed by using a previously described strategy[90]. Briefly, a Blasticidin-P2A-2×HA-FKBP12[F36V] cassette was integrated into the N-termini of endogenous *Wnk1* alleles in the MA9 leukaemia cell line. An sgRNA (sgWnk1_dTAG, see Supplementary Data 6) targeting a region close to the start codon of *Wnk1* was designed using the sgRNA designer tool provided by the Broad Institute (https://portals.broadinstitute.org/gppx/crispick/public) and commercially synthesised (Integrated DNA technologies UK). The 2 × HA-FKBP12[F36V] cassette flanked by 500 bp homology arms of the *Wnk1* gene was designed, synthesised, and cloned into a pUC19 plasmid as a donor template using the In-Fusion HD cloning kit (Takara, 638911). MA9 leukaemia cells were nucleofected with the sgRNA/Cas9 RNP complex and the donor plasmid using an SG Cell Line 4D X Kit S (Lonza, 197175). To prepare sgRNA/Cas9 RNP complex, 2 μl sgRNA (100 μM) was complexed with 2 μl Cas9 (45 μM) and incubated at room temperature (RT) for 20 min. 1 × 10[6] cells were harvested, washed with PBS, and centrifuged at 350 g for 3 min at RT. Pelleted cells were gently resuspended in 20 μl SG nucleofection buffer and then sgRNA/Cas9 RNP complex and 1 μg of the donor plasmid were added. Nucleofection was performed using the Lonza 4D-nucleofector X Unit (programme CP-100). Immediately after nucleofection, 200 μl of pre-warmed medium was added to the cuvette and the cell suspension was carefully transferred to 3 ml of pre-warmed medium in a 6-well plate. Cells were allowed to recover for 48 h at 37 °C before starting 10 μg/ml blasticidin selection of the pools. After 7 days of selection, surviving cells were single-cell sorted into 96-well plates using SONY MA900 cell sorter and maintained in medium containing 10 μg/ml blasticidin. After 14 days of culture, single-cell clones were split into pairs, with one receiving dTAG-13 treatment. Clones unable to survive the dTAG-13 treatment and their paired clones were then subject to additional screening for biallelic knock-in by PCR and verified by immunoblotting. Three clones were selected and kept for further experiments.

### CellTiter-Glo cell viability assay
Cells were plated in 96-well Nunc white microplates (Thermo Fisher Scientific, 136101) at a density of 1000 cells per well in a final volume of 100 μl medium. Compounds were added and incubated for the indicated duration. For the measurement of IC50 values, cells were incubated with the indicated compounds for 72 h. Assay plates were

removed from the incubator and allowed to equilibrate to RT before adding 100 μl of CellTiter-Glo reagent (Promega, G7570), according to the manufacturer's instructions. The assay plates were then shaken and incubated at RT on an orbital shaker for 10 min. Luminescence was measured using a GloMax Multiplus (Promega) or SpectraMax iD5 (Molecular Devices) plate reader. The GraphPad Prism software (v9) was used for plotting graphs and calculating IC50 values.

## Cell cycle analysis

Cells were treated with 500 nM OHT for the specified duration. EdU incorporation assays were conducted using the Click-iT™ EdU Alexa Fluor™ 488 Flow Cytometry Assay Kit (Thermo Fisher Scientific, C10425), following the manufacturer's instructions. Cells were pulsed with 1 μM EdU for 45 min prior to fixation. 4′,6-diamidino-2-phenylindole (DAPI) was used to co-stain the cells for measuring DNA content. Data was acquired on BD FACSAria III (BD Biosciences), and data analysis was performed with FlowJo software (v 10.8.1)

## Annexin V staining

Cells were treated with DMSO, EtOH, 10 μM Compound 12 or 500 nM OHT for the indicated time. A total of $2 \times 10^5$ cells were incubated with 3 μl APC Annexin V dye (BD Pharmingen™, 550474) for 30 min at RT, protected from light, and washed with 1× Annexin V Binding Buffer (BD Pharmingen™, 556454). Cells were resuspended in 150 μl of 1 × Annexin V Binding Buffer containing 20 ng/ml DAPI, and data was acquired using on CytoFLEX LX (Beckman). Data analyses were performed using FlowJo (v10.8.1).

## Animal studies

All mouse experiments conducted in Denmark were approved by the Danish Animal Ethics Committee under license number 2017-15-0201-01176. Similarly, all mouse experiments carried out in the UK were approved by the Animals in Science Regulation Unit (license number: PP5781054). All staff and animal technicians involved in the experiments possess the necessary accreditation to conduct animal experiments in the respective countries and are trained to maintain high standards of animal welfare. In both countries, a standard 12 h light–dark cycle was maintained, and mice were provided with unrestricted access to food and water. All animals were monitored daily and humanely euthanized at the experimental endpoint, which was not exceeded. In AML transplantation studies, the humane endpoint was determined by experimental design or, for survival studies, by disease progression, measured using a clinical scoring system outlined in the project licences. Parameters included weight, appearance, body condition, clinical signs, and behaviour.

## Generation of *Wnk1*$^{fl/+}$, *Wnk1*$^{fl/−}$, *Wnk1*$^{fl/D368A}$ MA9 leukaemia cells

*Wnk1*$^{fl/+}$, *Wnk1*$^{fl/−}$, *Wnk1*$^{fl/D368A}$ mice with a tamoxifen-inducible Cre recombinase in the *Rosa26* locus (*Rosa26-Cre*$^{ERT2}$) were previously generated[30]. Generation of MLL-AF9 transformed leukaemia was performed as previously described[43,91]. Briefly, bone marrow cells enriched for c-KIT$^+$ were obtained from *Wnk1*$^{fl/+}$, *Wnk1*$^{fl/−}$, *Wnk1*$^{fl/D368A}$ mice using CD117 MicroBeads (Miltenyi Biotec) and an autoMACS Pro cell separator (Miltenyi Biotec). These cells were transduced with MSCV-MLL-AF9-neo. Two days after transduction, cells were plated into semisolid methylcellulose medium (STEMcell technologies, MethoCult™ M3134) containing 50 ng/mL SCF, 10 ng/mL IL-6, 10 ng/mL IL-3 and 600 μg/ml neomycin for the first plating, followed by four rounds of consecutive replating. The pre-leukaemia cells were then transplanted into sublethally irradiated C57BL/6J mice through intravenous tail vein injection. Approximately 2 months after transplantation, primary leukaemia cells were harvested from the bone marrow and spleen of sick mice and cultured in the same medium as MA9 leukaemia cells described above.

## Bone marrow transplantation

$5 \times 10^4$ *Wnk1*$^{fl/+}$ or *Wnk1*$^{fl/−}$ or *Wnk1*$^{fl/D368A}$ leukaemia cells were injected into the tail vein of sublethally irradiated (650 rad) 6–8 week-old B6-SJL mice. Two weeks after transplantation, mice were treated with tamoxifen (75 mg/kg) for 7 consecutive days. The mice were monitored daily and humanely culled upon the display of AML symptoms. Kaplan–Meier survival analysis was performed using GraphPad Prism software (v9).

## In vivo Compound 12 treatment

To prepare Compound 12 for in vivo treatment, the compound was resuspended in sterile corn oil at 40 mg/ml. For in vivo treatment, $5 \times 10^4$ MA9 leukaemia cells were transplanted into sublethally irradiated (650 rad) 6–8 week-old B6-SJL mice via intravenous tail vein injection. Mice were treated intraperitoneally with either vehicle or 72 mg/kg/day of Compound 12 starting 4 days after transplantation for 14 consecutive days. The mice were monitored daily and humanely culled upon the display of AML symptoms. Kaplan–Meier survival analysis was performed using GraphPad Prism software (v9).

## Primary human AML patient samples and culture

Primary human AML samples were obtained from the Barts Cancer Institute Biobank. All samples were collected, and individual-level clinical data were shared and reported following informed consent and approval by the Barts Cancer Institute Ethical Committees and the BCI Tissue Biobank's scientific subcommittee, in compliance with the Declaration of Helsinki.

Frozen AML samples were retrieved from the Barts Cancer Institute Biobank. After thawing, *T* cells were depleted using the EasySep™ Human TCR Alpha/Beta Depletion Kit (Stem Cell Technologies, #17847). The enriched AML cells were then plated at a density of $0.4–1.0 \times 10^6$ mL in Myelocult H5100 medium (Stem Cell Technologies, #05150), supplemented with 20 ng/mL of IL-3 (#578004, BioLegend), G-CSF (#578604, BioLegend), and TPO (#763704, BioLegend). The cells were cultured either in liquid culture or in co-culture with MS-5 stromal cells.

IC50 analyses were performed after 72 h using the CellTiter-Glo assay in liquid culture. For WB analyses, primary patient samples were incubated for 2 h with the indicated concentration of WNK463 or vehicle control in liquid culture. Apoptosis was assessed using Annexin *V* staining g after 72 h of treatment with WNK463 or vehicle in co-culture with MS-5 stromal cells.

## Human primary AML xenotransplantation

For the xenograft models, eight- to ten-week-old NOD.Cg-*Kit*$^{W-41J}$*Tyr*$^+$*Prkdc*$^{scid}$ *Il2rg*$^{tm1Wjl}$/ThomJ (NBSGW) mice (Jackson Laboratory) were intravenously injected with $1 \times 10^6$ primary AML cells. Twelve weeks post-transplantation, the mice were treated for 14 consecutive days with either vehicle or WNK463 (1.5 mg/kg, administered orally every 12 h). Immediately following the treatment period, the mice were sacrificed. Human AML blast levels were evaluated by flow cytometric analysis of live hCD45 + hCD33 + hCD19-hCD3- cells in bone marrow, reflecting leukaemia progression. Mice were housed in well-ventilated cages under controlled environmental conditions in a pathogen-free facility.

## Measurement of protein synthesis

Protein synthesis was measured by O-Propargyl-puromycin (OP-Puro) incorporation, as previously described with modifications[92,93]. Briefly, $5 \times 10^5$ cells/ml leukaemia cells were plated in fresh complete culture medium prior to OP-Puro labelling. Cells were labelled with 10 μM OP-Puro (MedChem express, HY-15,680) for 30 min and then collected from wells and washed twice with cold PBS. Next, cells were fixed in 200 μl of 4% (wt/vol) PFA in PBS for 15 min at RT (20–25 °C). Cells were washed twice with PBS, then permeabilized in 200 μl PBS containing

0.1% saponin (Sigma, 47036) and 3% FBS (Gibco, 10500064) for 5 min at RT. For the azide-alkyne cycloaddition reaction, Alexa Fluor 555 conjugated with azide (Thermo Fisher Scientific, A20012) at 5 μM final concentration was used along with the Click-iT Cell Reaction Buffer Kit (Thermo Fisher Scientific, C10269). The reaction mixture was incubated at RT for 30 min, and then cells were washed twice with PBS and resuspended in 200 μl of PBS containing DAPI (1 μg/ml) and analysed by flow cytometry, as described previously[93].

### Immunoblotting and Immunoprecipitation
Cells were either counted and lysed in 1 × Laemmli sample buffer (LSB) (50 mM Tris-HCl, pH 6.8, 10% (v/v) glycerol, 4% (w/v) SDS, 0.1% (w/v) bromophenol blue, and 5% (v/v) β-mercaptoethanol) and boiled at 95 °C for 15 min or lysed in RIPA buffer (25 mM Tris-HCl pH 7.6, 150 mM NaCl, 1% NP-40, 1% sodium deoxycholate, 0.1% SDS) and protein concentration were determined using Bradford protein assays. The total cell lysates or protein extracts were loaded equally onto NuPAGE protein gels, run, and transferred to PVDF membranes. After blocking with 5% non-fat milk in TBS + 0.1% Tween-20, the membranes were probed with the indicated primary antibodies listed in Supplementary Data 7. HRP-conjugated secondary antibodies were used for staining, and signal detection was performed using Pierce™ ECL WB Substrate (Thermo Fisher Scientific, 32106) with a ChemiDoc MP Imaging System (Bio-Rad). For immunoprecipitation, HEK-293T cells transiently transfected with the indicated cDNA expression vectors were rinsed twice with ice-cold PBS and lysed in lysis buffer (50 mM Tris-HCl, pH 7.4, 150 mM NaCl, 2 mM MgCl2, 1% NP-40) with Halt™ Protease and Phosphatase Inhibitor Cocktail (Thermo Fisher Scientific, 78440). The supernatants of cell lysates were collected after centrifugation at 14,000 g at 4 °C. For immunoprecipitation, 30 μL of anti-HA magnetic beads (Thermo Fisher Scientific, 88836) or anti-V5 agarose affinity gels (Sigma, A7345) were added to each lysate and incubated with rotation for 4 h at 4 °C. Immunoprecipitates were washed four times with lysis buffer containing 150 mM NaCl. Immunoprecipitated proteins were eluted by the addition of 150 μl of 1 × LSB and incubation at RT for 30 min. Immunoprecipitated proteins were resolved and analysed by immunoblotting, as described above.

### Protein purification
The N-terminal fragment of human SLC38A2 fused to a C-terminal GST tag was expressed in BL21(DE3) pLysS E. coli. One liter of bacteria was grown at 37 °C, and when the optical density reached 0.6, the bacteria were treated with 0.5 mM IPTG and incubated overnight at 16 °C. Bacteria were pelleted by centrifugation at 4500 rpm for 15 min, resuspended in 20 ml lysis buffer (20 mM Tris-HCl pH 8.0, 300 mM NaCl, 5% glycerol) on ice, and homogenised by sonication (VCX 500, SONICS) at 35% amplitude for 3 s per round for a total of 3 min. The lysate was then centrifuged at 20,000 g for 1 h at 4 °C. The supernatant was filtered through 0.45 μM filter and incubated with the 3 ml Pierce Glutathione Spin Columns (Thermo Fisher Scientific, 16108) by rotating at 4 °C for 1 h. Proteins immobilised on the columns were washed and eluted into 3 ml elution buffer (125 mM Tris-HCl pH 8.0, 150 mM NaCl, 10 mM glutathione). The eluted proteins were stored at −80 °C after the addition of 10% glycerol.

### In vitro kinase assay
Recombinant OXSR1 (07-122) was purchased from Carna Biosciences. For the kinase assay, recombinant OXSR1 and SLC38A2 were incubated in a 20 μl reaction buffer (50 mM Tris-HCl, pH 7.5, 0.1 mM EGTA, 10 mM MgCl₂ and 1 mM DTT), along with 50 μM cold ATP and 2 μCi [γ−32P] ATP, for 1 h at 30 °C. After incubation, the reactions were terminated by boiling in LSB buffer and separated by 4–12% NuPAGE. The gels were dried, and radioactivity was determined by autoradiography using an Amersham typhoon IP phosphor imaging scanner.

### Targeted LC-MS metabolomics analysis
Metabolites were extracted from $5 \times 10^6$ cells/sample by adding 200 μl of ice-cold methanol to cell pellets in Eppendorf tubes and incubating them for 30 min on ice. The samples were then centrifuged at 3220 g for 10 min at 4 °C to separate the extract from the cell pellet. The resulting fresh extracts (30 μl/sample) were subjected to targeted, quantitative metabolomic analysis using the AbsoluteIDQ™ p180 kit (Biocrates Life Sciences AG, Innsbruck, Austria) by electrospray ionisation tandem MS. The samples were derivatised with phenylisothiocyanate and the metabolites were extracted and analysed according to the manufacturer's instructions. Sample analyses were performed on a Waters Acquity H-class UPLC coupled to a Xevo TQ-S triple-quadrupole MS/MS System (Waters Corporation, Manchester, UK). The metabolite concentrations in the cell extracts were measured using MassLynx™ and TargetLynx™ software (Waters, Manchester, UK) and the MetIDQ™ software package (Biocrates Life Sciences AG, Innsbruck, Austria). Quantification of the metabolites of each biological sample was achieved by reference to appropriate stable isotope internal standards. The method follows the United States Food and Drug Administration Guidelines "Guidance for Industry−Bioanalytical Method Validation (May 2001)", providing proof of reproducibility within a given error range.

### Measurement of intracellular glutamine/glutamate levels
Intracellular Glutamine/glutamate levels were measured using the Glutamine/Glutamate-Glo Assay kit (Promega, J8021) according to the manufacturer's instructions. Briefly, $4 \times 10^5$ cells/ml were incubated with fresh complete medium, along with the indicated concentrations of Compound12 or after OHT treatment. Cells were washed twice with cold PBS and then incubated with Inactivation Solution I (0.3 N HCl) for 5 min, followed by the addition of Tris Solution I (450 mM Tris, pH 8.0). No glutaminase enzyme solution was added to distinguish the glutamine and glutamate concentrations. The assay measured the total concentration of glutamine and glutamate. Glutamine/glutamate levels were normalised to the number of cells.

### Phosphoproteome analysis
**Sample preparation.** Proteins from 10 million cells per replicate were extracted using SEPOD method with modifications[94]. To avoid sonication, Benzonase was used to digest DNA. Since it is inactivated quickly in the SEPOD surfactant cocktail, cells were first resuspended in 125 μl ice-cold 0.1% SDS, 100 mM EPPS pH 8.5, with 1:100 phosphatase inhibitor cocktails 2 and 3 (Sigma, P5726 and P0044) and 1 cOmplete™ Mini EDTA-free Protease Inhibitor Cocktail tablet (Roche,11836170001) per 10 ml. After a 5 min incubation on ice, 2 × surfactant cocktail (300 mM NaCl, 4% SDS, 2% SDC, 4% NP40) was added to the samples and heated to 70 °C while shaking, which cleared the solution completely. The samples were then transferred to 96-well plates and the concentration was estimated by BCA assay to be 3.5–4 g/L. 250 μl of 20 mM TCEP and 20 mM CAA was added to each sample to reduce and alkylate proteins and decrease surfactant concentration prior to protein precipitation. A volume containing 400 μg of sample was taken to the next plate and precipitated in 2 ml protein LoBind plates (Eppendorf, C0030504305) for 1 h by the addition of 800 μl of −20 °C acetone. The samples were spun down at 4000 g for 30 min, washed twice with −20 °C acetone and digested in 250 μl of 10 mM EPPS by adding 4 μg of trypsin twice: 1) 4 h digestion at 37 °C; and 2) then overnight digestion at 25 °C (1:100). Digestion was performed by shaking at 37 °C in a thermoblock at 3000 rpm. After the digestion was complete, the concentration was around 1.6 g/L. 200 μg (125 μl) was labelled with 16 plex tags for 1 h in ACN as suggested by the manufacturer. The samples were not pulled at identical volumes, but the ratios of sample volumes were adjusted by mixing 1 μl of each fraction, running a 15 min gradient in with AIF[95] in duplicate, and then taking the

amount of sample that ensures the same total signal. 100 μg of the combined sample was fractionated with the High pH Reversed-Phase Peptide Fractionation Kit with the following steps (280 μl per fraction): 0, 7.5, 12.8, 14.8, 16.1, 17.1, 18.2, 19.2, 20.3, 21.5, 22.9, 26.3, 37, 50. The rest of the material was desalted on 300 mg of Oasis HLB material (Waters, 186007549), and the phosphopeptides were enriched using High-Select™ Fe-NTA, as suggested by the manufacturer. The enriched phosphopeptides were fractionated with High pH Reversed-Phase Peptide Fractionation Kit using the following steps: 0, 7.4, 9.2, 11.6, 13.8, 16.2, 50 (steps suggested as personal communication with the manufacturer).

**Data acquisition.** Evaporated samples were resuspended in 20 μl of 0.1%TFA, of which 5 μl was analysed by EASY-nLC 1200 System equipped with a 2 mm particle size, 75 mm × 500 mm easyspray column in direct injection mode. The samples were separated using a gradient of buffer A (0.1% formic acid in water) and buffer B (0.1%formic acid in acetonitrile) as follows: 0–7% for 5 min, 7–30% for 90 min, 30–50% for 20 min. The eluted peptides were analysed on Orbitrap Fusion Lumos Tribrid Mass Spectrometer using MS3 SPS (for not enriched samples) and MS3 MSA (for phospho-enriched samples) methods with the settings recommended by the instrument manufacturer, with the following modifications: 1) CID NCE for MS2 was set to 32; 2) HCD NCE for MS3 was set to 45; 3) $C$-series exclusion was disabled since TMTPro reagent was not enabled in $C$-series exclusion node.

**Data analysis.** Data were analysed using Proteome Discoverer software (v3.1). A database search was conducted with the Sequest HT search engine, using the Mouse UniProt database that only contained reviewed entries and canonical isoforms (retrieved on 10/10/2019). Variable modification of Oxidation (M) was set, while TMTPro was set as a fixed modification. A maximum of two missed cleavages were allowed. The precursor and fragment mass tolerances were set at 10 ppm and 0.6 Da, respectively. PSMs were validated by percolator with a 0.01 posterior error probability threshold. The PSM data was exported as .csv and rolled up to protein level using the MSstatsTMT package. Phosphosite quantitation data were normalised to protein level in $R$ and subjected to statistical tests in MSstatsTMT. The GO analysis was performed in clusterProfiler.

**Analysis of glutamine uptake with $^{15}N_2$-glutamine**
Cells were seeded onto 6-well plates at a density of 350,000 cells per well prior to extraction. Eight plates were used, and five wells per plate were used for metabolite extraction, with each well per plate representing a biological replicate. For the $^{15}N$-labelled glutamine tracing experiments, the medium was changed for the appropriate medium supplemented with 0.5 mM L-glutamine ($^{15}N2$, 98%) and 0.5 mM unlabelled glutamine at the start of the experiment. Half of the plates were supplemented with 10 μM Compound 12, while the other half were supplemented with an equivalent amount of DMSO. At each time point, the plates were placed on ice and the medium was removed carefully into labelled Eppendorf tubes placed on ice. The cells were then washed three times with ice-cold PBS before adding 1 ml of extraction buffer (acetonitrile/methanol/H2O, 40:40:20 v/v/v at −20 °C). Then, the cells were scraped and transferred into labelled Eppendorf vials. A 100 μl aliquot of the metabolite solution was then mixed with 100 μl of acetonitrile with 0.2 % acetic acid at −20 °C, and centrifuged for 10 min at 13,000 rpm at 4 °C. The supernatant was then transferred into an liquid chromatography–mass spectrometry (LC-MS) $V$-shape vials and 4 μl was injected into the LC-MS for analysis.

Aqueous normal phase liquid chromatography was performed using an Agilent 1290 Infinity II LC system equipped with a binary pump, a temperature-controlled auto-sampler (set at 4 °C) and a temperature-controlled column compartment (set at 25 °C) containing a cogent diamond hydride type $C$ silica column (150 × 2.1 mm; dead volume 315 μl). The flow rate was set at 0.4 ml/min. Polar metabolites were eluted using solvent $A$, which consisted of deionized water (resistivity -18 MW cm) and 0.2% acetic acid, and solvent $B$, which consisted of 0.2% acetic acid in acetonitrile. The gradient used was as follows: 0 min 85% B; 0–2 min 85% B; 3–5 min to 80% B; 6–7 min 75% B; 8–9 min 70% B; 10–11 min 50% B; 11.1–14 min 20% B; 14.1–25 min hold 20% B followed by a 5 min re-equilibration period at 85% B at a flow rate of 0.4 ml/min. Accurate mass spectrometry was carried out using an Agilent Accurate Mass 6545 Q-TOF instrument. Dynamic mass axis calibration was achieved by continuous infusion, post-chromatography, of a reference mass solution using an isocratic pump connected to an ESI ionisation source operated in the positive-ion mode. The nozzle voltage and fragmentor voltage were set at 2000 and 100 V, respectively. The nebuliser pressure was set at 50 psig, and the nitrogen drying gas flow rate was set at 5 L/min. The drying gas temperature was maintained at 300 °C. The MS acquisition rate was 1.5 spectra/s, and m/z data ranging from 50–1200 were stored. This instrument enabled accurate mass spectral measurements with an error of less than 5 parts-per-million (ppm), mass resolution ranging from 10,000–45,000 over the m/z range of 121–955 atomic mass units, and a 100,000-fold dynamic range with picomolar sensitivity. The data were collected in the centroid 4 GHz (extended dynamic range) mode. Detected m/z were deemed to be identified metabolites based on unique accurate mass-retention time and MS/MS fragmentation identifiers for masses exhibiting the expected distribution of accompanying isotopomers. Under these experimental conditions, typical variation in abundance for most of the metabolites remained between five and 10%.

For labelling experiments, the fractional enrichment for each metabolite was determined by dividing the peak height ion intensities of each labelled species by the ion intensities of both labelled and unlabelled species using the software Agilent Profinder version B.8.0.00 service pack 3. For total metabolite levels, peak height ion intensities for each metabolite were determined using the software Agilent Profinder version B.8.0.00 and normalised to total protein levels measured via a BCA assay following the manufacturer's instructions. Media metabolite levels were determined by subtracting the blank media peak height ion intensities for each metabolite from each sample's peak height ion intensities and dividing by the blank peak height ion intensities for each metabolite. Negative values represent net metabolite consumption from the media, while positive values represent net secretion into the media.

**Statistical analysis**
Two-sided Student's $t$ tests were used for significance testing unless stated otherwise. Significance levels are indicated as follows: *$p < 0.05$, **$p < 0.01$, ***$p < 0.001$ and ****$p < 0.0001$. Graphs and error bars represent means ± SD of independent biological experiments unless specified otherwise.

**Reporting summary**
Further information on research design is available in the Nature Portfolio Reporting Summary linked to this article.

# Data availability
Proteomics data are available via ProteomeXchange with identifier PXD041902. Source data are provided with this paper.

# Code availability
There is no custom code in this study, only publicly available tools were used in data analysis as described wherever relevant in the methods section.

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

## Acknowledgements

We thank all members of the Helin lab for helpful discussions. J.-E.M. was supported by the Novo Nordisk Foundation (NNF) Copenhagen Bioscience PhD Programme (NNF18CC0033666). P.G. was supported by a Cancer Research UK Advanced Clinician Scientist fellowship C57799/A27964. The work in the Helin Lab was supported by the Danish Cancer Society (R167-A10877), the Kirsten and Freddy Johansen's Research Prise, the Novo Nordisk Foundation to the NNF Centre for Stem Cell Biology (no. NNF17CC0027852), the Danish Research Centre for Precision Medicine in Blood Cancers funded by the Danish Cancer Society Grant number R223-A13071 the Greater Copenhagen Health Science Partners, and The Institute of Cancer Research. We wish to thank the Barts Cancer Institute Haemato-Oncology tissue bank for sample collection and processing and anonymised data sharing. We also wish to thank the patients who have generously donated their tissues and shared their data to be used in the generation of this publication.

## Author contributions

S.D. performed most of the presented experiments with help from K.A., J.E.M. and X.H. K.A. generated the mouse sgRNA library and performed the sgRNA screens. K.A., K.N. and I.P.-R. performed the in vivo experiments. I.P.-R. and S.D. performed the mouse PDX experiments, and S.D. processed the samples and analysed the data. S.D. performed apoptosis analysis with help from H.D. P.S. performed mass spectrometry analysis, supervised by R.C.H. S.D. and P.S. analysed the MS data. S.D. and M.K. performed the glutamine uptake assay, supervised by G.P. S.D. performed the in vitro kinase assay, assisted by M.D. in data analysis and interpretation. S.D., A.P. and Y.A. performed targeted LC-MS metabolomics analysis, supervised by F.I.R. A.V.D. and R.R. synthesised Compound 12. R.K. and V.L.J.T. provided conditional *Wnk1* mice. P.G. provided advice on human primary AML xenotransplantation and facilitated the experiments. S.D. and K.H. wrote the manuscript, and all authors commented on the manuscript. K.H. supervised the overall study and acquired funding.

## Competing interests

K.H. and K.A. are co-founders of Dania Therapeutics. K.H. is a scientific advisor for Hannibal Innovation and was recently a scientific advisor for Inthera Bioscience AG and for MetaboMed Inc. The remaining authors declare no competing interests.
