## [Transparent peer review file · Nature Communications]

WNK1 signalling regulates amino acid transport and mTORC1 activity to sustain AML growth

Corresponding Author: Professor Kristian Helin

Version 0:

Reviewer comments:

Reviewer #1

(Remarks to the Author)

In this work the WNK1-OSR1/STK39 pathway is identified as a critical, previously uncharacterised dependency in AML. Genetic depletion and pharmacological inhibition of WNK1 or its downstream targets (OSR1 and STK39) significantly reduce cell proliferation and induce apoptosis in leukaemia cells both in vitro and in vivo. This pathway regulates mTORC1 signalling by controlling amino acid uptake through the phosphorylation of amino acid transporters such as SLC38A2.

The study highlights the potential of targeting the WNK1-OSR1/STK39 pathway for AML treatment and suggests that inhibitors of this pathway could be effective across various cancer types.

The work is original in the context of AML. WNK1 has been demonstrated as a potential target in multiple myeloma, as shown in <https://doi.org/10.1016/j.celrep.2024.114211> and in T-ALL <https://doi.org/10.1182/blood-2022-163069>. However, a unifying mechanism of action has not been recognized, suggesting that further work is required to dissect this pathway.

Please find below the reviewer suggestions or request:

Line 137. Per policy I believe that data can not be omitted and should be clearly presented

Line 142. Protein expression analysis for OSR1/STK39 in cell line with and without knock down should be reported
Figure 2e legends are distorted

Line 145-152. It is clear that the validation attempt is guided by the presence of an MA9 model. In this section, it was decided to validate the WNK1 effector OSR1 and not STK39 because it is not expressed in the model. The relevance in the human AML is unclear. If both proteins are expressed in human and if both need to be suppressed to observe a phenotype once WNK1 is knocked down, it is possible that the catalytic function of OSR1 is irrelevant because it is compensated by STK39. A series of experiments is necessary, at least in human cell lines, to determine if knockdown of WNK1 cause an effect on both OSR1 and STK39 proteins. Protein should be presented for all the experiments with genetic modulation.

Line 166. Please change the sentence in a less definitive statement. Only two proteins have been validated, based on work reported from the literature.

Figure 3: very elegant experiments

Figure 4: We understand that both WNK463 and Compound 12 are tool compounds. However, I think it is inappropriate to call models sensitive if they respond only to concentrations greater than 10 μ M after six days in culture. What happens to the levels of p-proteins at 12, 24, 36 hours, etc.? Is there a rebound in p-levels observed? Please complete p-proteomic tracking upon WNK463 and Compound 12 tested.

Figure 5. Previous studies have highlighted that bumetanide modifies p-WNK1 levels. The authors do not show the levels of p-WNK1 (activating and inhibitory sites) and how their modulation is impacted by treatment with WNK463, Compound 12, and bumetanide. This is a clear shortcoming of the work that needs to be addressed. The lack of effect on NKCC1 and NKCC2 by bumetanide cannot be taken as justification without verifying that the drug does not modify p-WNK1 levels. The effect of WNK1 on mTOR is widely predictable from previous studies and therefore diminishes the novelty impact brought by this section of the work. Furthermore, although it is possible to demonstrate an anti-leukemic effect in vitro, their role in clinical practice is very limited even in a series of combination trials.

Figure 6. Bone marrow microenvironment is relative hypoxic microenvironment all these experiments in vitro in cells derived from mouse may reflect observation driven by the model and they can not be generalized. Experiment in primary sample

must be provided

In general, the study is professionally prepared and contains information with a certain degree of novelty. The technological effort appears excessive for validation on mTORC and amino acid metabolism. The study lacks validation in human models outside of cell lines. If the aim is to demonstrate a generic message on WNK1, human samples must be used and they should be representative of recurrent genetic lesions. The levels of p-WNK1 need to be analyzed, and synergy studies conducted with mTORC pathway inhibitors.

Reviewer #2

(Remarks to the Author)

Through a focused CRISPR screen, Duan et al. identify WNK1 as a dependency of AML cells. Based on this, authors conduct extensive genetic and pharmacological experiments to demonstrate that targeting WNK1 or its downstream effectors, OXSR1/STK39 suppresses AML cell viability and growth. The authors then examine the underlying mechanism and provide evidence that WNK1 inhibition leads to decreased amino acid uptake and resulting decrease in mTORC1 activity. This is an interesting paper that provides convincing evidence concerning the role of WNK1 for AML growth, both in vitro and in mouse models. The results may point to a new strategy for treating cancer. They also potentially have implications for the understanding of how nutrient uptake and mTORC1 signaling are regulated. The flow of experiments is logical and easy to follow, and the authors discuss results and open questions in a balanced manner. The later part of the study linking WNK1-OXSR1/STK39 to amino acid transporters and mTORC1 signaling reports striking phenotypes but is mechanistically somewhat preliminary. This part would benefit from additional experiments to strengthen the model proposed by the authors.

Major points:

1. Throughout the manuscript, the authors conclude that WNK1 inhibition mainly acts by suppressing mTORC1, but very few experiments compare the effects of inhibiting WNK1/OXSR1/STK39 or mTORC1. To support their conclusion, the authors should compare the effect of inhibiting either kinase on protein synthesis, cell viability and growth directly. As AML cells seem to be partially rapamycin-insensitive, it might be helpful to include an mTOR kinase inhibitor.
2. The data linking WNK1 to amino acid uptake and mTORC1 activation are not very convincing. Most amino acids that have been identified as direct mTORC1 activators (Leu, Arg, Met) are not decreased by WNK1 suppression. The exception is Gln, which activates mTORC1 in a Rag-independent manner (PMID: 25567907). However, NPRL2 KO rescues mTORC1 activity upon WNK1 inhibition, suggesting that this effect depends on the GATOR-Rag pathway. To clarify the effect of WNK1 inhibition on mTORC1, it would be helpful to conduct mTORC1 re-activation experiments, including with single amino acids – e.g. known mTORC1 activators, amino acids, whose intracellular levels decrease upon WNK1 inhibition. WNK1 inhibition causes dramatic cell death. Can the authors exclude that mTORC1 activity is suppressed because cells are dying, rather than due to a specific decrease in amino acid uptake?
3. The rescue of mTORC1 activity by NPRL2 KO in Compound 12-treated cells is clear, but does this also rescue cell viability/growth?
4. The data concerning OXSR1/STK39-mediated regulation of amino acid transporters are rather vague. Please show the phosphorylation change of the transporters for the proteomics data (Fig5b). The inhibition of amino acid transporters is a key aspect of the mechanism suggested by the authors for growth inhibition by WNK1. I understand that it is difficult to study amino acid transporters genetically due to functional redundancy (although co-deletion of two putative WNK1-regulated transporters might have an effect). However, SLC38A2 deletion has a modest effect on cell growth (ED Fig. 8f), and it would be helpful to know whether this can be rescued by expression of phosphomutant vs wt variants.

Minor points:

1. Fig. 1 b-d and similar experiments. Are the cell competition assays based on single measurements? The authors show this effect for multiple murine and human cell lines, so this is not a major concern. However, it would be helpful to specify this in figure legends.
2. WNK1 knockout does not suppress cell proliferation in MEFs (Fig. 1c), suggesting a selective effect in cancer cells. However, MEFs are highly proliferative and it is unclear why the suggested mechanism of action – suppression of amino acid uptake and mTORC1 signaling – would not operate in this cell type. It would be helpful to know whether WNK1 KO affects mTORC1 signaling in MEFs.
3. Statistical analysis is not meaningful for technical replicates (e.g. Fig. 4b-d), as these measurements are not independent.
4. Fig. 5b: The mTORC1 phosphotargets shown for the proteomics experiment are rather selective. To assess these results, it would be helpful to include a larger number of known targets.

Reviewer #3

(Remarks to the Author)

The submission by Duan and colleagues presents cell biologic, biochemical, genetic, and preclinical studies assessing WNK1 and its downstream kinase effectors OXSR1 and STK39 as candidate dependencies in AML. WNK1 emerged from a targeted in vitro CRISPR screen. Strengths of this work include the range of experimental techniques utilized, interesting biochemical and proteomic data regarding WNK1 in TORC1 signaling and amino acid sensing, and the expertise of the lab in this area. However, in my opinion, the data presented do not support the author's contention that WNK1 is "critical" (abstract) or "essential" for AML growth (e.g., lines 103, 104). Based on the data presented, I am skeptical that WNK1 represents a promising therapeutic target in AML. Specific comments follow below.

(1) Key data presented in Figure 1 and elsewhere in the manuscript are based on competitive fitness assays, which are well-established for comparing the relative growth of isogenic (or near isogenic) matched cell populations but do not test essentiality. Published work has shown that the competitive fitness of mouse and human leukemia cells that relapse after responding to drug treatment is frequently reduced in comparison to the primary leukemia. Yet they are aggressive enough to kill the patient (or mouse). Establishing that a specific gene is essential for cancer growth is challenging and approaches like those shown in Figure 3 where the growth of specific cell populations are assessed singly are required. The claim of essentiality is further undermined by the data shown in Figure 3e-3g where WNK1 ablation extended the survival of recipient mice by about 2-fold but did not cure them and of the preclinical trial in Fig. 4g. Along these lines, it would be of interest to know if genotyping was performed at the *Wnk1* locus when the mice shown in Figs 3e-3g were euthanized to assess if *Wnk1* function is required for disease progression as stated in line 193.

(2) The authors used an established model of AML driven by retroviral expression of MLL-AF9 followed by extensive selection by serial replating in methylcellulose. These cells and similar models developed by Zuber and other are tractable for performing genetic screens but are highly sensitized to perturbations and less flexible/resilient than primary AMLs. Given this and the subsequent use of the MA9 model for much of the data presented, I have concerns regarding the relevance of this work beyond MLL/KMT2A rearranged leukemia. The authors might clarify this point as they seem to take a different view.

(3) Like most commercial AML cell lines, THP1 (NRAS), NOMO1 (KRAS), HL60 (NRAS), U937 (PTPN11), MV4-11 (FLT3), and MOLM-13 (FLT3) harbor signaling mutations. Acknowledging the limitations of using competitive growth assays noted above, it is striking that the two FLT3-mutant lines show better fitness than the other 4. This is of potential interest given the author's proteomic data implicating WNK1 in mTORC signaling. Is it possible that mutant FLT3 sustains TORC activation when WNK1 is knocked out or inhibited chemically?

(4) A very minor point – the authors might insert "in days" after "over time" in the Fig. 1 legend to indicate what "D" refers to in the figure panels.

(5) With the caveat that the labels for panels "e" and "j" were distorted in the printed version of the MS that I read, the authors might address a few questions/comments about Figure 2: (1) it might be of interest to ask if MA9 cells grown under non-competitive conditions can remain viable and grow out by up-regulating STK39 after OXSR1 knockdown; (2) are the "AML cells" noted in the Figure legend for panel "d" THP-1, which would make sense as THP-1 cells are shown in panel "e"; and, (3) why were THP-1 cells used in panel "e" and NOMO1 cells in panel "e"?

(6) While Fig. 2f is excellent, I think breaking the long sentence that starts on line 145 into two – one for each protein – could enhance understanding.

(7) In Figure 4, it would be ideal to show similar data for THP1 as those presented in Fig. 4b for MA9 cells. Line 205 notes decreased proliferation in human and mouse AML cell lines. How many human cell lines in addition to THP1 were tested? Based on the data in Figure 1, testing MOLM13 would be of particular interest as these cells appear to be less dependent on WNK1 than other human AML cell lines.

(8) The data in Figs 4c, 4d suggest that both inhibitors are cytostatic rather than cytotoxic. The authors might comment on this.

(9) Given the excitement around menin inhibitors as a mechanism-based therapies for MLL-rearranged leukemias, it seems important to compare the IC50 values of compound 12 and WNK463 to one of these clinical compounds given the author's optimism regarding WNK1 as a promising therapeutic target.

(10) The data presented in Fig. 4G are unimpressive although statistically significant (this is largely due to the extremely predictable time of death of the control mice). An increase in survival of <2-fold in a trial of this nature is very unlikely to predict efficacy. The authors seem to attribute this to the poor drug-like properties of Compound 12 but present no data that directly address this.

(11) The proteomic and biochemical studies presented in Figures 5 and 6 are well done – I found this to be the most interesting part of the paper. With this said, I found the paragraph that starts on line 299 to be overly long and little distracting. Some of this background might be summarized in a simple figure panel.

(12) The second paragraph of the Discussion is a little disappointing. The 3 papers the authors reference as showing "great promise" for inhibiting TORC1 in AML were published in 2005-2012. They do acknowledge the clinical failure of rapamycin and other inhibitors in AML, which has led many in the field and in the biopharma industry to move on to other more

promising targets. I am also a little perplexed that the authors seem to imply that inhibiting TORC1 activation by WNK1 will not result in feedback reactivation. I suggest the authors consider a more balanced presentation of this area and the challenges in credentialing/restoring TORC1 as a therapeutic target in AML.

Version 1:

Reviewer comments:

Reviewer #1

(Remarks to the Author)

The authors have comprehensively addressed the requests for clarification by adding a significant number of experiments supporting the hypothesis that WNK1 is a mediator of mTOR. The overall lack of studies in PDX models reduces the enthusiasm, as does the apparent overfitting on mTOR in cells that are largely undergoing apoptosis (e.g., Fig. 5b, 48h WNK f/-). This aspect should be further discussed in the conclusions, particularly in light of the substantial failure of mTOR inhibitors in this context

Reviewer #2

(Remarks to the Author)

The authors have addressed my concerns. The new results corroborate the mechanism through which WNK1 regulates amino acid transport and mTORC1 signaling, and the manuscript will be of broad interest to researchers interested in these signaling pathways and their relevance for AML.

Reviewer #3

(Remarks to the Author)

I previously reviewed this submission by Duan and colleagues and also read the incisive comments of the other two reviewers. My expertise is primarily in AML biology and therapeutics rather than in metabolism/amino acid uptake and how this is regulated by mTORC. The authors have generated considerable new data that are presented in the resubmission. As a non-expert, I remain most impressed by the studies of the role of WNK1 and its effector kinases in regulating TORC1/amino acid transporters. I am decidedly less convinced that targeting WNT1 will be efficacious in AML (i.e., have a beneficial therapeutic index in vivo in primary leukemia cells). With this said, I don't think additional experimentation would change the central conclusions of this work in a meaningful way and therefore support publication if the other two reviewers are in agreement. I would ask the authors consider adopting a more conservative tone in some areas as noted below.

Lines 46-47. While I agree with the statement about WNK1 signaling playing a "critical" role in amino acid uptake, I don't think this applies to AML progression. My operating definition of having a "critical" role in AML or any other cancer is a protein that when biochemically inhibited by a small molecule results in objective regression of the cancer (not delayed regrowth) and the development of intrinsic genetic or epigenetic resistance at relapse. Examples of this in AML are "on target" BCR-ABL, FLT3, or menin mutations that confer resistance after inhibitor treatment and differentiation with secondary RAS mutations after venetoclax. The data presented in this paper fall very far short of this bar.

Lines 48-49. I would delete this sentence as there are no data presented in this paper in non-AML cancers and I don't know of any cancers where mTORC1 inhibitors are the standard of care.

Line 181-182. I would interpret the data shown in Figures 3e-3g as showing that loss of WNK1 and its catalytic activity impair the growth of AML cells in vivo.

Lines 217-224. The effects of WNK263 treatment in reducing AML burden was not statistically significant. Given this, the last two sentences in that paragraph are not supported by the data presented and I suggest deleting them.

Lines 381-382. As I mentioned in my previous review, the 3 citations about leukemia are from over a decade ago because mTOR inhibition was ineffective and has largely been abandoned as a therapeutic strategy. I didn't see any hard data cited that this situation is different now. So perhaps the authors might consider adding that mTOR remains a "theoretically" appealing target?

Detailed Response to Reviewers' Comments

Reviewer #1

In this work the WNK1-OSR1/STK39 pathway is identified as a critical, previously uncharacterised dependency in AML. Genetic depletion and pharmacological inhibition of WNK1 or its downstream targets (OSR1 and STK39) significantly reduce cell proliferation and induce apoptosis in leukaemia cells both in vitro and in vivo. This pathway regulates mTORC1 signalling by controlling amino acid uptake through the phosphorylation of amino acid transporters such as SLC38A2.

The study highlights the potential of targeting the WNK1-OSR1/STK39 pathway for AML treatment and suggests that inhibitors of this pathway could be effective across various cancer types.

The work is original in the context of AML. WNK1 has been demonstrated as a potential target in multiple myeloma, as shown in <https://doi.org/10.1016/j.celrep.2024.114211> and in T-ALL <https://doi.org/10.1182/blood-2022-163069>. However, a unifying mechanism of action has not been recognized, suggesting that further work is required to dissect this pathway.

Please find below the reviewer suggestions or request:

Line 137. Per policy I believe that data can not be omitted and should be clearly presented

The reviewer is correct, and we have now included the data (Extended Data Fig. 2a) showing the expression of *Oxsr1* and *Stk39* in MA9 cells, with MEF cells used as a positive control.

Line 142. Protein expression analysis for OXSR1/STK39 in cell line with and without knock down should be reported

In addition to the THP1 cells, we have now included additional human cell lines to demonstrate that OXSR1/STK39 expression was efficiently targeted by the indicated sgRNAs (Extended Data Fig. 2b and c). As shown in Fig. 2d and Extended Data Fig. 2b and c, the sgRNAs for OXSR1/STK39 effectively targeted OXSR1/STK39 in Cas9-expressing human AML cells. It should be noted that the sgRNAs for OXSR1 were cloned into a GFP-based reporter vector, while the sgRNAs for STK39 were cloned into an RFP-based reporter vector. Therefore, for the competition assays, the relative proportions of GFP-positive and RFP-positive cells were measured.

Figure 2e legends are distorted

We have corrected the distorted legends.

Line 145-152. It is clear that the validation attempt is guided by the presence of an MA9 model. In this section, it was decided to validate the WNK1 effector OSXR1 and not STK39 because it is not expressed in the model. The relevance in the human AML is unclear. If both proteins are expressed in human and if both need to be

suppressed to observe a phenotype once WNK1 is knocked down, it is possible that the catalytic function of OXSR1 is irrelevant because it is compensated by STK39. A series of experiments is necessary, at least in human cell lines, to determine if knockdown of WNK1 causes an effect on both OXSR1 and STK39 proteins. Protein should be presented for all the experiments with genetic modulation.

We thank the reviewer for these comments. We have now included data in a new Extended Figure 3a, showing that deletion of *WNK1* does not result in changes of OXSR1 and STK39 protein levels in human leukaemia cells. Moreover, we agree that the catalytic function of OXSR1 is not essential in cells where both OXSR1 and STK39 are expressed. This is illustrated in Figure 2 d, e and Extended Data Figure 2b-h), showing that both OXSR1 and STK39 need to be deleted to see an effect on human AML proliferation. These results agree with previous studies suggesting that OXSR1 and STK39 are functionally redundant kinases¹⁻³.

This interpretation is further supported in human AML cells by showing that ectopic expression of either the constitutively active phosphomimetic mutant of OXSR1 (OXSR1^{T185E,S325E}) or the constitutively active phosphomimetic mutant of STK39 (STK39^{T233E,S373E}) restored the proliferation defects caused by the disruption of endogenous WNK1. In contrast, neither OXSR1^{WT} nor STK39^{WT} could rescue the defect (Fig. 2j and Extended Data Fig. 2i,j).

For clarity, we have revised the conclusion highlighted by the Reviewer to: "These results demonstrate that the catalytic activity of OXSR1 is required for cell proliferation in MA9 cells lacking STK39 expression, and that the phosphomimetic OXSR1 mutant retains its catalytic activity," emphasizing the importance of OXSR1's catalytic function in MA9 cells lacking STK39 expression.

Line 166. Please change the sentence to a less definitive statement. Only two proteins have been validated, based on work reported from the literature.

We have revised the conclusion to: 'Taken together, these results suggest that the role of WNK1 in AML growth is mediated, at least in part, through the phosphorylation of OXSR1/STK39, as supported by this and previous studies.'

Figure 3: very elegant experiments

We thank the reviewer for the positive feedback on Figure 3.

Figure 4: We understand that both WNK463 and Compound 12 are tool compounds. However, I think it is inappropriate to call models sensitive if they respond only to concentrations greater than 10 μ M after six days in culture. What happens to the levels of p-proteins at 12, 24, 36 hours, etc.? Is there a rebound in p-levels observed? Please complete p-proteomic tracking upon WNK463 and Compound 12 tested.

As suggested by the reviewer, we have now included data assessing phospho-OXSR1 and phospho-STK39 levels following treatment with WNK1 inhibitors in a time-dependent manner. As shown in Extended Data Fig. 3l, the levels of phospho-OXSR1 and phospho-STK39 remained largely stable over a 72-hour period. However, a slight rebound was observed after 12 hours, which was accompanied by

a significant increase in the total protein levels of OXSR1 and STK39. Therefore, it appears the cells try to compensate for the lack of WNK1 mediated phosphorylation of OXSR1 and STK39 by increasing the expression levels of the proteins.

Figure 5. Previous studies have highlighted that bumetanide modifies p-WNK1 levels. The authors do not show the levels of p-WNK1 (activating and inhibitory sites) and how their modulation is impacted by treatment with WNK463, Compound 12, and bumetanide. This is a clear shortcoming of the work that needs to be addressed. The lack of effect on NKCC1 and NKCC2 by bumetanide cannot be taken as justification without verifying that the drug does not modify p-WNK1 levels. The effect of WNK1 on mTOR is widely predictable from previous studies and therefore diminishes the novelty impact brought by this section of the work. Furthermore, although it is possible to demonstrate an anti-leukemic effect in vitro, their role in clinical practice is very limited even in a series of combination trials.

We thank the reviewer for this important comment. To the best of our knowledge, there are no reports demonstrating that bumetanide modifies phospho-WNK1 levels. One previous study demonstrated that bumetanide did not lead to WNK1 activation⁴. In our study, we used a concentration of bumetanide higher than the 10 μ M concentration known to affect NKCC1/2 activity in HEK293 cells⁵. However, we acknowledge that even higher concentrations might not be effective in inhibiting NKCC1/2, and that we do not provide a positive control showing that NKCC1/2 are inhibited. Given that this is not the primary focus of our study and in the interest of caution, we believe it would be prudent to remove the data related to bumetanide from the manuscript, which we have done in the revised version of the manuscript.

While previous studies have implicated WNK1 in regulating processes like autophagy, direct evidence linking WNK1 to mTORC1 regulation remains limited. A recent study demonstrated that WNK1 influences autophagy—a process also regulated by mTORC1—suggesting a potential indirect connection between WNK1 and mTORC1 in this context⁶. However, our study provides direct evidence that WNK1 regulates mTORC1, specifically through its role in regulating amino acid transport. This novel insight into the mechanistic interaction between WNK1 and mTORC1 contributes to a deeper understanding of the mTORC1 pathway, which is crucial in cancer biology. Given this, our findings could inform the development of more targeted and effective cancer treatments, particularly through combinatory strategies targeting WNK1 and the two downstream kinases OXSR1 and STK39 together with mTORC1 inhibitors.

Figure 6. Bone marrow microenvironment is relative hypoxic microenvironment all these experiments in vitro in cells derived from mouse may reflect observation driven by the model and they can not be generalized. Experiment in primary sample must be provided

Yes, we agree that this could strengthen our results, and in the revised version of the manuscript we have incorporated data from five patient-derived primary AML samples, including both MLL-rearranged (MLL-r) and non-MLL-r subtypes.

Our results demonstrate that WNK1 inhibition with WNK463 consistently reduces p-OXSR1/p-STK39 and p-P70S6K levels (with the exception of AML#1, as noted) and significantly increases apoptosis in treated samples compared to controls (Fig.4h-j).

These data demonstrate that WNK1 inhibition is effective not only in mouse-derived and human cell line models but also in primary AML samples (not previously expanded in tissue culture), supporting the translational relevance of our observations.

In general, the study is professionally prepared and contains information with a certain degree of novelty. The technological effort appears excessive for validation on mTORC and amino acid metabolism. The study lacks validation in human models outside of cell lines. If the aim is to demonstrate a generic message on WNNK1, human samples must be used and they should be representative of recurrent genetic lesions. The levels of p-WNK1 need to be analyzed, and synergy studies conducted with mTORC pathway inhibitors.

We sincerely appreciate the reviewer's comments. Regarding the analysis of p-WNK1 and primary samples, we have addressed these points in detail in our responses to Figures 5 and 6 (please refer to the responses above).

To explore the potential synergy between WNK1 and mTORC1 pathway inhibition, we treated leukaemia cells with WNK463 in combination with two mTORC1 inhibitors: rapamycin and Torin1 (a kinase inhibitor of mTOR). As shown in Response Figure 1, co-inhibition of WNK1 and mTORC1 demonstrated synergistic effects in leukaemia cells. While these findings are promising, further investigation is required to elucidate the underlying mechanisms and evaluate whether combination therapy with WNK1 and mTORC1 inhibitors could represent a potential therapeutic strategy for AML. We believe this important question extends beyond the scope of the current study and will be addressed in future work.

Response Fig.1

Response Fig.1 The Highest Single Agent (HSA) synergy score distribution map, generated using the SynergyFinder online tool, illustrates the synergistic effects of WNK463 in combination with rapamycin (upper panel) or Torin1 (lower panel). HSA scores above 10 indicate synergistic effects. The concentrations used were as follows: WNK463 (0, 1, 3, and 10 μM), rapamycin (0, 0.0001, 0.0003, and 0.001 μM), and Torin1 (0.003, 0.01, and 0.03 μM).

In response to the reviewer's suggestion regarding additional human models, we utilized a human primary xenograft model. Eight- to ten-week-old NBSGW mice were intravenously injected with 1×10^6 primary AML cells. Twelve weeks post-transplantation, the mice were treated for 14 consecutive days with either vehicle or WNK463 (1.5 mg/kg, administered orally, twice daily). The mice were sacrificed

immediately after the treatment period and the human AML blast levels were assessed by flow cytometry, measuring hCD45⁺/hCD33⁺ cells in the bone marrow as an indicator of leukaemia progression.

As shown in Extended Data Fig. 5a,b, the results reveal a strong trend toward reduced leukaemia progression in WNK463-treated mice compared to the vehicle group. However, the high biological variability in the engraftment of the human primary AML samples, and the presence of two outliers in the WNK463-treated group limits our ability to draw definitive conclusions from this dataset. It is important to notice that we have used primary human AML samples in our studies, rather than established human PDX cells, which are commonly used in these types of studies, and we consider them similar to established human AML cell lines.

Reviewer #2

Through a focused CRISPR screen, Duan et al. identify WNK1 as a dependency of AML cells. Based on this, authors conduct extensive genetic and pharmacological experiments to demonstrate that targeting WNK1 or its downstream effectors, OXSR1/STK39 suppresses AML cell viability and growth. The authors then examine the underlying mechanism and provide evidence that WNK1 inhibition leads to decreased amino acid uptake and resulting decrease in mTORC1 activity. This is an interesting paper that provides convincing evidence concerning the role of WNK1 for AML growth, both in vitro and in mouse models. The results may point to a new strategy for treating cancer. They also potentially have implications for the understanding of how nutrient uptake and mTORC1 signaling are regulated. The flow of experiments is logical and easy to follow, and the authors discuss results and open questions in a balanced manner. The later part of the study linking WNK1-OXSR1/STK39 to amino acid transporters and mTORC1 signaling reports striking phenotypes but is mechanistically somewhat preliminary. This part would benefit from additional experiments to strengthen the model proposed by the authors.

Major points:

1. Throughout the manuscript, the authors conclude that WNK1 inhibition mainly acts by suppressing mTORC1, but very few experiments compare the effects of inhibiting WNK1/OXSR1/STK39 or mTORC1. To support their conclusion, the authors should compare the effect of inhibiting either kinase on protein synthesis, cell viability and growth directly. As AML cells seem to be partially rapamycin-insensitive, it might be helpful to include an mTOR kinase inhibitor.

It is well established that mTORC1 plays a crucial role in regulating protein synthesis, cell viability, and growth⁷. In line with the reviewer's suggestion, we have conducted additional experiments and now provide new data demonstrating that treatment with mTOR inhibitors results in a significant reduction in protein synthesis and cell viability, as well as a marked inhibition of cell growth in leukaemia cells (Extended Data Fig. 7b, e, f).

2. The data linking WNK1 to amino acid uptake and mTORC1 activation are not very

convincing. Most amino acids that have been identified as direct mTORC1 activators (Leu, Arg, Met) are not decreased by WNK1 suppression. The exception is Gln, which activates mTORC1 in a Rag-independent manner (PMID: 25567907). However, NPRL2 KO rescues mTORC1 activity upon WNK1 inhibition, suggesting that this effect depends on the GATOR-Rag pathway. To clarify the effect of WNK1 inhibition on mTORC1, it would be helpful to conduct mTORC1 re-activation experiments, including with single amino acids – e.g. known mTORC1 activators, amino acids, whose intracellular levels decrease upon WNK1 inhibition. WNK1 inhibition causes dramatic cell death. Can the authors exclude that mTORC1 activity is suppressed because cells are dying, rather than due to a specific decrease in amino acid uptake?

We thank the reviewer for the thoughtful and insightful comments. It is important to stress that knockout of NPRL2 only partially rescues mTORC1 activity upon WNK1 inhibition (Fig. 6a), suggesting that Rag GTPase-independent mechanisms may also be involved. Furthermore, as shown in Fig. 6b, WNK1 inhibition leads to a decrease in intracellular levels of Gln as well as Leu, along with other amino acids whose roles in mTORC1 activation have not been well characterised.

Following the reviewer's suggestion, we conducted an mTORC1 re-activation experiment after amino acid starvation, using Leu as a known mTORC1 activator. As shown in Extended Data Fig. 8g, mTORC1 activity was rescued by adding Leu or Gln following amino acid starvation in WNK1-uninhibited cells. However, in WNK1-inhibited cells, the addition of Leu or Gln failed to restore mTORC1 activity. This result indicates that WNK1 inhibition affects the uptake of these amino acids into the cells, thereby preventing the restoration of mTORC1 activity, and/or that WNK1 inhibition, in addition to affecting amino acid uptake, also affects the activity of other proteins regulating mTORC1 activity.

The immediate effect (within in 30 min) of WNK1 inhibition on amino acid uptake, mTORC activity and proteins synthesis (e.g. Figure 5f, 6e, and Extended Data Fig. 6a), strongly suggest that mTORC1 suppression is a direct and immediate consequence of WNK1 inhibition rather than a secondary or indirect effect through a cellular phenotype, such as cell death or cell cycle arrest. To further investigate this, we assessed cell death through Annexin V staining in a time-course experiment in response to WNK1 inhibition. As shown in Extended Data Fig. 6b, no significant increase in apoptotic cells was observed following 4 hours of WNK1 inhibition.

3. The rescue of mTORC1 activity by NPRL2 KO in Compound 12-treated cells is clear, but does this also rescue cell viability/growth?

NPRL2 depletion did not rescue cell growth following WNK1 depletion (Extended Data Fig. 8f), suggesting that NPRL2 depletion can restore mTORC1 signalling but cannot compensate for the amino acid deficiency that impairs cell proliferation upon WNK1 inhibition/depletion.

4. The data concerning OXSR1/STK39-mediated regulation of amino acid transporters are rather vague. Please show the phosphorylation change of the transporters for the proteomics data (Fig5b). The inhibition of amino acid transporters is a key aspect of the mechanism suggested by the authors for growth

inhibition by WNK1. I understand that it is difficult to study amino acid transporters genetically due to functional redundancy (although co-deletion of two putative WNK1-regulated transporters might have an effect). However, SLC38A2 deletion has a modest effect on cell growth (ED Fig. 8f), and it would be helpful to know whether this can be rescued by expression of phosphomutant vs wt variants.

We have now included the regulated phosphopeptides of the amino acid transporters in Fig. 5b. We acknowledge the limitations of our study by not establishing how phosphorylation affects transporter activity. In fact, we have worked extensively on identifying the regulatory phosphorylation sites on SLC38A2 as also described in the manuscript; however, due to the high number of serine and threonine residues in the N-terminal of this transmembrane protein (12 serines and 4 threonines within the N-terminal 75 amino acids), pinpointing specific sites has proven challenging. Additionally, at least four amino acid transporters exhibited significantly reduced phosphorylation, making it difficult to delete them all to assess their collective impact.

Our results show that WNK1 is required for amino acid uptake, mTOR activation and protein synthesis. Moreover, we show that the phosphorylation of several amino acid transporters is decreased in response to WNK1 inhibition. Based on these results, we suggest that the decrease in amino acid uptake is due to lower activity of the amino acid transporters, however, since we observe a decrease in phosphorylation of several of the amino acid transporters, and they are functionally redundant, it has so far not been possible to establish the precise role of the phosphorylation of the amino acid transporters.

Minor points:

1. Fig. 1 b-d and similar experiments. Are the cell competition assays based on single measurements? The authors show this effect for multiple murine and human cell lines, so this is not a major concern. However, it would be helpful to specify this in figure legends.

Yes, most of the competition assays are based on single measurements; however, they have been confirmed or validated in different cell lines, and some have been repeated at least twice. We have now specified this in the respective figure legends.

2. WNK1 knockout does not suppress cell proliferation in MEFs (Fig. 1c), suggesting a selective effect in cancer cells. However, MEFs are highly proliferative and it is unclear why the suggested mechanism of action – suppression of amino acid uptake and mTORC1 signaling – would not operate in this cell type. It would be helpful to know whether WNK1 KO affects mTORC1 signaling in MEFs.

That is a good question. As shown in Extended Data Fig.6c, genetic knockout of WNK1 in MEFs does not appear to significantly affect mTORC1 activity. We have not investigated the underlying reasons for this further, however, it could be due to redundant functions of other WNK family proteins in MEFs that compensate for the loss of WNK1.

3. Statistical analysis is not meaningful for technical replicates (e.g. Fig. 4b-d), as these measurements are not independent.

Thanks for pointing this out. We agree that technical replicates are not independent measurements, and their statistical analysis should be interpreted with caution. To strengthen the results, we have included additional data using an additional cell line, as well as supplementary growth curve analyses (in revised Fig. 4b-d and Extended Data Fig. 3h-k).

4. Fig. 5b: The mTORC1 phosphotargets shown for the proteomics experiment are rather selective. To assess these results, it would be helpful to include a larger number of known targets.

The data in Fig. 5b were initially intended to highlight the top significant mTORC1 targets affected by WNK1 inhibition. In response to the reviewer's suggestion, we have now included additional known mTORC1 targets and amino acid transporters in the revised Fig. 5b.

Reviewer #3

The submission by Duan and colleagues presents cell biologic, biochemical, genetic, and preclinical studies assessing WNK1 and its downstream kinase effectors OXSR1 and STK39 as candidate dependencies in AML. WNK1 emerged from a targeted in vitro CRISPR screen. Strengths of this work include the range of experimental techniques utilized, interesting biochemical and proteomic data regarding WNK1 in TORC1 signaling and amino acid sensing, and the expertise of the lab in this area. However, in my opinion, the data presented do not support the author's contention that WNK1 is "critical" (abstract) or "essential" for AML growth (e.g., lines 103, 104). Based on the data presented, I am skeptical that WNK1 represents a promising therapeutic target in AML. Specific comments follow below.

(1) Key data presented in Figure 1 and elsewhere in the manuscript are based on competitive fitness assays, which are well-established for comparing the relative growth of isogenic (or near isogenic) matched cell populations but do not test essentiality. Published work has shown that the competitive fitness of mouse and human leukemia cells that relapse after responding to drug treatment is frequently reduced in comparison to the primary leukemia. Yet they are aggressive enough to kill the patient (or mouse). Establishing that a specific gene is essential for cancer growth is challenging and approaches like those shown in Figure 3 where the growth of specific cell populations are assessed singly are required. The claim of essentiality is further undermined by the data shown in Figure 3e-3g where WNK1 ablation extended the survival of recipient mice by about 2-fold but did not cure them and of the preclinical trial in Fig. 4g. Along these lines, it would be of interest to know if genotyping was performed at the *Wnk1* locus when the mice shown in Figs 3e-3g were euthanized to assess if *Wnk1* function is required for disease progression as stated in line 193.

We appreciate the reviewer's thoughtful comments. For the specific question, we did not do the genotyping of the *Wnk1* locus in the mice shown in Figs 3e-3g, and we

therefore performed an experiment similar to the one presented in Fig. 3f (Extended Data Fig. 3f). As shown in Extended Data Fig. 3f,g, and in agreement with the data presented in Fig. 3f, the deletion of *Wnk1* significantly extended the lifespan of the transplanted mice, however, it did not cure the mice. To establish whether the AML cells in the euthanized sick mice expressed *Wnk1*, we genotyped the cells, which showed that they still contained the wild-type *Wnk1* locus. This result indicates that WNK1 is required for leukaemia growth in vivo and is consistent with the in vitro data showing that surviving leukaemia cells from OHT-treated *Wnk1^{f/-}* cultures were *Wnk1*-non-floxed cells (Extended Data Fig. 3e).

(2) The authors used an established model of AML driven by retroviral expression of MLL-AF9 followed by extensive selection by serial replating in methylcellulose. These cells and similar models developed by Zuber and other are tractable for performing genetic screens but are highly sensitized to perturbations and less flexible/resilient than primary AMLs. Given this and the subsequent use of the MA9 model for much of the data presented, I have concerns regarding the relevance of this work beyond MLL/KMT2A rearranged leukemia. The authors might clarify this point as they seem to take a different view.

Excellent comments. We initially used the MLL-AF9 transformed leukaemia model for screening due to its advantages, such as rapid disease onset, amenability to genetic manipulation, and the aggressive nature of this model. However, we recognize the limitations of this model. After validating our findings in this model, we extended our analysis to human cell lines, including both MLL-rearranged (MLL-r) and non-MLL-rearranged (non-MLL-r) subtypes (Fig. 1d and Fig. 4e). Additionally, we have now included data using primary human AML samples, which demonstrate that WNK1 inhibition significantly reduced the growth of primary AML cells (Fig.4h-j).

(3) Like most commercial AML cell lines, THP1 (NRAS), NOMO1 (KRAS), HL60 (NRAS), U937 (PTPN11), MV4-11 (FLT3), and MOLM-13 (FLT3) harbor signaling mutations. Acknowledging the limitations of using competitive growth assays noted above, it is striking that the two FLT3-mutant lines show better fitness than the other 4. This is of potential interest given the author's proteomic data implicating WNK1 in mTORC signaling. Is it possible that mutant FLT3 sustains TORC activation when WNK1 is knocked out or inhibited chemically?

This is an interesting question, which we had not thought about. However, as shown in Response Fig.2 below, there does not appear to be a correlation between FLT3 mutation status and the response to WNK1 inhibition with respect to mTORC1 signalling, acknowledging the small number of samples we have tested.

Response Fig.2

Response Fig.2 Immunoblot analysis of the indicated proteins in human cell lines treated with Compound 12 (C12) (10 μM, 2 h)

(4) A very might minor point – the authors might insert “in days” after “over time” in the Fig. 1 legend to indicate what “D” refers to in the figure panels.

We have inserted “in days (D)” after “over time” in the relevant figure legends.

(5) With the caveat that the labels for panels “e” and “j” were distorted in the printed version of the MS that I read, the authors might address a few questions/comments about Figure 2:

We apologize for the distorted labels in panels e and j of Figure 2. These labels have now been corrected in the revised manuscript.

1) it might be of interest to ask if MA9 cells grown under non-competitive conditions can remain viable and grow out by up-regulating STK39 after OXR1 knockdown;

Thank you for the interesting comments. Since STK39 is not expressed in MA9 cells (Extended Data Fig. 2a) and OXR1 is required for MA9 cell growth, we first established stable Cas9⁺ MA9 cells overexpressing human STK39. We then knocked out endogenous OXR1 using GFP-tagged OXR1 sgRNAs and sorted GFP-positive cells to isolate OXR1 knockout populations. As shown in **Response Fig. 3a and 3b**, the cells remain viable and proliferate following OXR1 knockdown. These findings are consistent with the results in human cells (Fig. 2e and Extended Data Fig. 2b-h), demonstrating that STK39 and OXR1 have redundant functions.

Response Fig.3

Response Fig.3 Immunoblot analysis (a) and growth curve (b) of Cas9⁺ MA9 leukemia cells stably expressing human STK39, transduced with either a non-targeting sgRNA (NCtrl) or *Oxsr1*-targeting sgRNAs.

2) are the “AML cells” noted in the Figure legend for panel “d” THP-1, which would makes sense as THP-1 cells are shown in panel “e”; and,

We thank the reviewer for identifying this inconsistency. The “AML cells” noted in the figure legend for panel d are indeed THP-1 cells, consistent with panel e. We apologize for the confusion and have deleted “and NOMO1” from the figure legend to clarify.

3) why were THP-1 cells used in panel “e” and NOMO1 cells in panel “e”?

THP-1 cells were used in panel e, while NOMO1 cells were moved to Extended Data Fig. 2e due to space limitations. We regret not updating the figure legend to reflect this change and have now corrected it accordingly.

(6) While Fig. 2f is excellent, I think breaking the long sentence that starts on line 145 into were two – one for each protein – could enhance understanding.

We have revised the sentence to enhance clarity:

“WNK1 activates OXSR1 by phosphorylating a conserved threonine residue (Thr185) within its catalytic T-loop domain and a serine residue (Ser325) located within the non-catalytic C-terminal domain. Similarly, WNK1 activates STK39 by phosphorylating a conserved threonine residue (Thr233) within its catalytic T-loop domain and a serine residue (Ser373) located within the non-catalytic C-terminal domain.”

(7) In Figure 4, it would be ideal to show similar data for THP1 as those presented in Fig. 4b for MA9 cells. Line 205 notes decreased proliferation in human and mouse AML cell lines. How many human cell lines in addition to THP1 were tested? Based on the data in Figure 1, testing MOLM13 would be of particular interest as these cells appear to be less dependent on WNK1 than other human AML cell lines.

We have performed the experiments suggested by the reviewer and have now replaced the former Fig. 4c,d with growth curve data for THP-1 and MOLM13 cells treated with Compound 12 at 1, 3, and 10 μ M. We also include growth curve data for THP-1 and MOLM13 cells treated with WNK463 at 1, 3, and 10 μ M (Extended Data Fig. 3h,i). In addition to several non-AML cell lines, eight human AML cell lines were tested with WNK1 inhibitors, as shown in Fig. 4e. Moreover, we have tested the

effect of WNK1 inhibition on five primary AML patient samples, now included in the revised manuscript (Fig.4h-j).

(8) The data in Figs 4c, 4d suggest that both inhibitors are cytostatic rather than cytotoxic. The authors might comment on this.

We agree with the reviewer that the data in those figure panels suggest that. However, our results suggest that WNK1 inhibition both leads to cytostatic (Fig 4b-d; Extended Data Figs. 3h-k) and cell death (Figs 3b,c; 4j; Extended Data Figs 3c; 6d; 7c; 8f) in the tested AML cells. Whereas the deletion of Wnk1 in the MA9 cells is highly cytotoxic, the inhibition of WNK1 using small molecule inhibitors leads both to cytostatic and cytotoxic effects. Although, we know that cytostatic and cytotoxic effects in cells are strongly affected by the genotypes of the cells, our results suggest that the cytotoxic effects are primarily observed when WNK1 activity is completely inhibited (deletion) and in sensitive cells (primary human AML cells), whereas the cytostatic effect is more pronounced in the established human AML cell lines using the current available inhibitors to WNK1.

(9) Given the excitement around menin inhibitors as a mechanism-based therapies for MLL-rearranged leukemias, it seems important to compare the IC50 values of compound 12 and WNK463 to one of these clinical compounds given the author's optimism regarding WNK1 as a promising therapeutic target.

That is indeed an interesting suggestion. However, Menin-MLL interaction inhibitors have undergone extensive optimization and clinical validation. For example, in murine MLL-AF9 transformed leukaemia cells, the IC50 values for first-generation inhibitors MI-2 and MI-3 were approximately 5 μM ⁸, while the second-generation inhibitors MI-503 and MI-463 demonstrate significantly improved potency, with IC50 values of 0.22 μM and 0.23 μM , respectively⁹. The third-generation inhibitor VTP50469 further advances the field with an IC50 of 15nM¹⁰. These inhibitors have shown remarkable progress in targeting MLL-rearranged and NPM1-mutated AML, and the recent FDA approval of Revumenib for relapsed or refractory acute leukemia with KMT2A rearrangements marks a significant milestone in this field.

Unfortunately, the current available WNK1 inhibitors, Compound 12 and WNK463 have IC50 values in the μM range, similar to the first-generation Menin inhibitors. Therefore comparing, the current WNK1 inhibitors to VTP50469, the approved Menin inhibitor does not give valuable insights regarding the potential of targeting WNK1 in AML. In fact, we propose that the development of more potent WNK1 inhibitors to clarify whether WNK1 is a suitable therapeutic target for the treatment of AML.

(10) The data presented in Fig. 4G are unimpressive although statistically significant (this is largely due to the extremely predictable time of death of the control mice). An increase in survival of <2-fold in a trial of this nature is very unlikely to predict efficacy. The authors seem to attribute this to the poor drug-like properties of Compound 12 but present no data that directly address this.

We acknowledge that the increase in survival observed in this experiment, while statistically significant, is relatively modest. However, we believe that the observed increase in survival, though not dramatic, still provides valuable information about

the potential of targeting WNK1 in AML using a tool compound small-molecule inhibitor, and they support the data shown in Figure 3e-g. We would also like to refer to studies that tested the efficacy of inhibiting BRD4¹¹ and BCL2¹² in preclinical xenograft models, where the degree of life extension was similar to the one, we show in Figure 4g. The BCL2 inhibitor, Venetoclax has subsequently been approved for the treatment of AML whereas the BRD4 inhibitors are still in clinical trials.

(11) The proteomic and biochemical studies presented in Figures 5 and 6 are well done – I found this to be the most interesting part of the paper. With this said, I found the paragraph that starts on line 299 to be overly long and little distracting. Some of this background might be summarized in a simple figure panel.

We thank the reviewer for the positive feedback and appreciate the suggestions regarding the paragraph starting on line 299. In response, we have amended the text and included a summary in a new figure panel in Extended Data Fig. 7g.

(12) The second paragraph of the Discussion is a little disappointing. The 3 papers the authors reference as showing “great promise” for inhibiting TORC1 in AML were published in 2005-2012. They do acknowledge the clinical failure of rapamycin and other inhibitors in AML, which has led many in the field and in the biopharma industry to move on to other more promising targets. I am also a little perplexed that the authors seem to imply that inhibiting TORC1 activation by WNK1 will not result in feedback reactivation. I suggest the authors consider a more balanced presentation of this area and the challenges in credentialing/restoring TORC1 as a therapeutic target in AML.

We have revised the discussion in response to the reviewer’s comment, and hope the reviewer finds the discussion more balanced.

1. Parrish, P.C.R. *et al.* Discovery of synthetic lethal and tumor suppressor paralog pairs in the human genome. *Cell Reports* **36** (2021).
2. Vitari, A.C., Deak, M., Morrice, N.A. & Alessi, D.R. The WNK1 and WNK4 protein kinases that are mutated in Gordon's hypertension syndrome phosphorylate and activate SPAK and OSR1 protein kinases. *Biochemical Journal* **391**, 17-24 (2005).
3. Moriguchi, T. *et al.* WNK1 regulates phosphorylation of cation-chloride-coupled cotransporters via the STE20-related kinases, SPAK and OSR1. *Journal of Biological Chemistry* **280**, 42685-42693 (2005).
4. Zagorska, A. *et al.* Regulation of activity and localization of the WNK1 protein kinase by hyperosmotic stress. *J Cell Biol* **176**, 89-100 (2007).
5. Boyd-Shiwarski, C.R. *et al.* WNK kinases sense molecular crowding and rescue cell volume via phase separation. *Cell* **185**, 4488-4506 e4420 (2022).
6. Gallolu Kankanamalage, S. *et al.* Multistep regulation of autophagy by WNK1. *Proc Natl Acad Sci U S A* **113**, 14342-14347 (2016).
7. Liu, G.Y. & Sabatini, D.M. mTOR at the nexus of nutrition, growth, ageing and disease. *Nat Rev Mol Cell Bio* **21**, 183-203 (2020).
8. Grembecka, J. *et al.* Menin-MLL inhibitors reverse oncogenic activity of MLL fusion proteins in leukemia. *Nat Chem Biol* **8**, 277-284 (2012).

9. Borkin, D. *et al.* Pharmacologic inhibition of the Menin-MLL interaction blocks progression of MLL leukemia in vivo. *Cancer Cell* **27**, 589-602 (2015).
10. Krivtsov, A.V. *et al.* A Menin-MLL Inhibitor Induces Specific Chromatin Changes and Eradicates Disease in Models of MLL-Rearranged Leukemia. *Cancer Cell* **36**, 660-673 e611 (2019).
11. Dawson, M.A. *et al.* Inhibition of BET recruitment to chromatin as an effective treatment for MLL-fusion leukaemia. *Nature* **478**, 529-533 (2011).
12. Pan, R. *et al.* Selective BCL-2 inhibition by ABT-199 causes on-target cell death in acute myeloid leukemia. *Cancer Discov* **4**, 362-375 (2014).

Detailed Response to Reviewers' Comments

Reviewer #1:

The authors have comprehensively addressed the requests for clarification by adding a significant number of experiments supporting the hypothesis that WNK1 is a mediator of mTOR. The overall lack of studies in PDX models reduces the enthusiasm, as does the apparent overfitting on mTOR in cells that are largely undergoing apoptosis (e.g., Fig. 5b, 48h WNK f⁻). This aspect should be further discussed in the conclusions, particularly in light of the substantial failure of mTOR inhibitors in this context

We thank the reviewer for the positive assessment of our revised manuscript.

In our previous response to Reviewer #2's comment, "Can the authors exclude that mTORC1 activity is suppressed because cells are dying?", we stated:

"The immediate effect (within 30 minutes) of WNK1 inhibition on amino acid uptake, mTORC1 activity, and protein synthesis (e.g., Figures 5f, 6e, and Extended Data Fig. 6a) strongly suggests that mTORC1 suppression is a direct and immediate consequence of WNK1 inhibition, rather than a secondary or indirect effect due to cellular phenotypes such as apoptosis or cell cycle arrest. To further investigate this, we assessed cell death using Annexin V staining in a time-course experiment following WNK1 inhibition. As shown in Extended Data Fig. 6b, no significant increase in apoptotic cells was observed within 4 hours of WNK1 inhibition."

Regarding Fig. 5b, cells were treated with the WNK1 inhibitor Compound 12 for 3 hours, not 48h, during which we did not observe a significant increase in apoptosis.

We agree with the reviewer that the limited testing of WNK1 inhibitors in a broader panel of PDX models is a limitation, and we have acknowledged this in the revised discussion.

The so-called 'clinical failure' of mTOR inhibitors is a worthwhile discussion. We would say it depends on the expectations. Yes, there have been many clinical trials that have not given a positive outcome, however, everolimus (a Rapamycin analogue) has been approved for the treatment for various cancers, including kidney cancer, metastatic pancreatic neuroendocrine tumours, giant cell astrocytoma, and GI neuroendocrine tumours. Moreover, there are currently 150 ongoing clinical trials with everolimus. Therefore, we do not believe that mTOR inhibitors have failed, but it would be fairer to say that for many they have not lived up to expectations.

The lack of positive response in clinical trials could be due to the limited efficacy of the rapalog-class of mTOR inhibitors as monotherapies. Moreover, as noted in our previous response, we observed a strong synergistic effect when combining WNK1 and mTOR inhibitors. This finding suggests that dual inhibition of these pathways may represent a promising therapeutic strategy, warranting further investigation in both preclinical and translational settings.

Reviewer #2:

The authors have addressed my concerns. The new results corroborate the mechanism through which WNK1 regulates amino acid transport and mTORC1 signaling, and the manuscript will be of broad interest to researchers interested in these signaling pathways and their relevance for AML.

We thank the reviewer for the positive assessment of our revised manuscript.

Reviewer #3:

I previously reviewed this submission by Duan and colleagues and also read the incisive comments of the other two reviewers. My expertise is primarily in AML biology and therapeutics rather than in metabolism/amino acid uptake and how this is regulated by mTORC. The authors have generated considerable new data that are presented in the resubmission. As a non-expert, I remain most impressed by the studies of the role of WNK1 and its effector kinases in regulating TORC1/amino acid transporters. I am decidedly less convinced that targeting WNT1 will be efficacious in AML (i.e., have a beneficial therapeutic index in vivo in primary leukemia cells). With this said, I don't think additional experimentation would change the central conclusions of this work in a meaningful way and therefore support publication if the other two reviewers are in agreement. I would ask the authors consider adopting a more conservative tone in some areas as noted below.

Lines 46-47. While I agree with the statement about WNK1 signaling playing a "critical" role in amino acid uptake, I don't think this applies to AML progression. My operating definition of having a "critical" role in AML or any other cancer is a protein that when biochemically inhibited by a small molecule results in objective regression of the cancer (not delayed regrowth) and the development of intrinsic genetic or epigenetic resistance at relapse. Examples of this in AML are "on target" BCR-ABL, FLT3, or menin mutations that confer resistance after inhibitor treatment and differentiation with secondary RAS mutations after venetoclax. The data presented in this paper fall very far short of this bar.

We have revised the conclusion to: "Our findings underscore an important role of the WNK1-OXSR1/STK39 pathway in regulating amino acid uptake and driving AML progression."

Lines 48-49. I would delete this sentence as there are no data presented in this paper in non-AML cancers and I don't know of any cancers where mTORC1 inhibitors are the standard of care.

As the reviewer suggested, we removed the following sentence: 'Moreover, our mode-of-action studies suggest that WNK1 and OXSR1/STK39 inhibitors could be effective in a broad range of cancer types.'

Line 181-182. I would interpret the data shown in Figures 3e-3g as showing that loss of WNK1 and its catalytic activity impair the growth of AML cells in vivo.

As the reviewer suggested, we have revised the conclusion to “Taken together, these results demonstrate that loss of WNK1 and its catalytic activity impairs AML cell growth in vivo.”

Lines 217-224. The effects of WNK263 treatment in reducing AML burden was not statistically significant. Given this, the last two sentences in that paragraph are not supported by the data presented and I suggest deleting them.

As the reviewer suggested, we removed the following sentence: “Nevertheless, a clear trend indicated that the human AML burden was lower in WNK463-treated mice. Therefore, taken together, our results suggest that WNK1 inhibitors could be used as an effective therapeutic strategy in AML.”

Lines 381-382. As I mentioned in my previous review, the 3 citations about leukemia are from over a decade ago because mTOR inhibition was ineffective and has largely been abandoned as a therapeutic strategy. I didn't see any hard data cited that this situation is different now. So perhaps the authors might consider adding that mTOR remains a “theoretically” appealing target?

We understand the reviewer and many others have been disappointed with the clinical efficacy of mTOR inhibitors. This does not mean mTOR inhibition and inhibition of the mTOR pathway is ‘theoretically’ appealing, because everolimus has in fact been approved for the treatment of a series of different cancers. Moreover, there are more than 150 ongoing clinical trials using everolimus (RAD001) for the treatment of cancer. Everolimus has not been approved for the treatment of AML, however, we still think that the mTOR pathway is an appealing target, which is what is stated in the sentence.